

# CALIOPE-Urban v1.0: Coupling R-LINE with a mesoscale air quality modelling system for urban air quality forecasts over Barcelona city (Spain)

Jaime Benavides[1], Michelle Snyder[2], Marc Guevara[1], Albert Soret[1], Carlos Pérez García-Pando[1], Fulvio Amato[3], Xavier Querol[3], and Oriol Jorba[1]

[1]Barcelona Supercomputing Center, Spain
[2]Institute for the Environment, University of North Carolina at Chapel Hill, USA
[3]Institute of Environmental Assessment and Water Research, IDAEA-CSIC, Spain

*Correspondence to:* Oriol Jorba (oriol.jorba@bsc.es)

**Abstract.** The $NO_2$ annual air quality limit value is systematically exceeded in many European cities. In this context, understanding human exposure, improving policy and planning, and providing forecasts requires the development of accurate air quality models at urban (street–level) scale. We describe CALIOPE-Urban, a system coupling CALIOPE - an operational mesoscale air quality forecast system based on HERMES (emissions), WRF (meteorology) and CMAQ (chemistry) mod-

els - with the urban roadway dispersion model R-LINE. Our developments have focused on Barcelona city (Spain), but the methodology may be replicated for other cities in the future. WRF drives pollutant dispersion and CMAQ provides background concentrations to R-LINE. Key features of our system include the adaptation of R-LINE to street canyons, the use of a new methodology that considers upwind grid cells in CMAQ to avoid double counting traffic emissions, a new method to estimate local surface roughness within street canyons, and a vertical mixing parametrization that considers urban geometry

and atmospheric stability to calculate surface level background concentrations. We show that the latter is critical to correct the nighttime overestimations in our system. Both CALIOPE and CALIOPE-Urban are evaluated using two sets of observations. The temporal variability is evaluated against measurements from five traffic sites and one urban background site for April-May 2013. While both systems show a fairly good agreement at the urban background site, CALIOPE-Urban shows a better agreement in traffic sites. The spatial variability is evaluated using 182 passive dosimeters that were distributed across Barcelona

during two weeks for February-March 2017. In this case, also the coupled system shows a more realistic distribution than the mesoscale system, which systematically underpredicts $NO_2$ close to traffic emission sources. Overall CALIOPE-Urban improves mesoscale model results, demonstrating that the combination of both scales provides a more realistic representation of $NO_2$ spatio-temporal variability in Barcelona.

## 1 Introduction

Persistent exposure to high $NO_2$ atmospheric concentrations in cities causes detrimental health effects (e.g., Sunyer et al., 2015; Barone-Adesi et al., 2015). In 2016, 19 out of the 28 European Union (EU) countries reported $NO_2$ exceedances of



the annual air quality limit value (40 μg m$^{-3}$) mostly at urban traffic monitoring stations (EEA, 2018). In this context there is a need for NO$_2$ data at street level in urban areas that enables individuals and communities to mitigate the problem by, for example, walking in less polluted streets or reducing traffic in school areas. However, both the poor density of air quality monitoring stations and the resolution of mesoscale air quality modeling systems (on the order of 1-km grid resolution), do
not adequately represent the NO$_2$ concentration gradients that typically occur near heavily trafficked streets (Duyzer et al., 2015; Borge et al., 2014). Urban dispersion models can estimate these gradients but their use has been typically limited to historic periods partly because the needed background concentrations and meteorological input have been approximated using observations (Vardoulakis et al., 2003).

In order to overcome these limitations, coupling regional and urban scale models has been recently found to be successful
in some cities. Hood et al. (2018) coupled a regional climate-chemistry model with 5 km horizontal resolution (EMEP4UK) with the fine-scale model ADMS-URBAN to simulate air quality over London in 2012. They compared the coupled system results with the regional and the fine-scale models run separately. Authors found that both the fine-scale model and the coupled system performed better than the regional for NO$_2$ at both annual mean and hourly concentration levels due to the explicit treatment of traffic emissions within the city. Jensen et al. (2017) estimated annual NO$_2$ concentrations at 2.4 million addresses
in Denmark using the street canyon model OSPM coupled with DEHM for regional background concentrations and UBM for urban background obtaining a good correlation in Copenhaguen ($r^2$ = 0.70) against 98 measurement sites for NO$_2$ in the year 2012. In addition, Maiheu et al. (2017) estimated EU-wide NO$_2$ annual average levels at 100 meter resolution with a regional model coupled with a dispersion kernel-based method. The approach does not produce hourly concentration levels and approximates road-link level traffic emissions by distributing the regional grid cell traffic emissions to each road-link based on
road capacity. Hence, it provides more spatial detail than previous EU scale NO$_2$ assessment studies, but more specific methods are required to resolve air quality in cities. In this sense, there is a lack of air quality urban forecasting methodologies that can be applied to a diverse range of cities and that consistently resolve at least some of the major challenges already identified by the community, i.e., 1) downscaling regional meteorology to street level as required to drive pollutant dispersion; 2) obtaining background concentrations from the mesoscale system avoiding the double counting of traffic emissions. Additionally, we
consider vertical mixing with background air a key process to be resolved when coupling the regional and urban scales.

Different approaches to downscale mesoscale meteorology are found in the research literature. Brousse et al. (2016) applied the Weather Research and Forecasting meteorological model (WRF) using the Building Effect Parametrization (Martilli et al., 2002) over Madrid considering WUDAPT Local Climate Zone data (Bechtel et al., 2015). This approach increases the mesoscale model's ability to resolve urban processes but does not reproduce the specific meteorological conditions in each
street as required by dispersion models. Kochanski et al. (2015) used a simplified CFD (QUIC) in combination with WRF to estimate wind conditions at street level. Hood et al. (2018) estimate an urban canopy flow field at the same resolution of the regional model. This calculation is based on the variation of surface roughness within the city. This approach includes the variation of some atmospheric stability parameters (e.g. friction velocity) but it neglects the variation of vertical mixing with background air depending on atmospheric stability and urban geometry. On the other hand, Jensen et al. (2017) do not consider
atmospheric stability within the street canyon model OSPM and within the vertical mixing with background air. The approach





presented here to downscale mesoscale meteorology to street-scale describing wind conditions and atmospheric stability in each street can be a promising solution to drive dispersion models and vertical mixing.

Background concentrations can be obtained from observations or mesoscale models, which are commonly used in forecasting applications. However, coupling mesoscale and urban dispersion models can lead to a double counting of traffic emissions.

To avoid double counting, Arunachalam et al. (2014) multiply urban background site observations by an estimated ratio between two mesoscale air quality simulations. The first run contains all the emission sources and the second neglects traffic emissions. Lefebvre et al. (2011) and Stocker et al. (2014) run first the urban dispersion model at mesoscale grid resolution with only traffic emissions and subtract its result to the mesoscale model simulation, which includes all the emission sources. Then, street scale model outputs are added to the result from the prior computation at finer resolution. Although, these methods

avoid double counting emissions they do not explicitly account for vertical mixing, a process that occurs at the intersection of regional and street scales. Urban air quality models such as SIRANE (Soulhac et al., 2011) have already implemented vertical mixing depending on local meteorology. In this study, we will show that this process may be relevant and explain some systematic errors found in the literature: nighttime $NO_2$ concentration values tend to be overestimated and afternoon values tend to be underestimated in traffic areas (e.g., Hood et al., 2018). Further efforts are necessary to explicitly resolve processes

happening among scales and to correct these biases in the mentioned periods of the day.

This work describes a methodology to couple the mesoscale air quality forecasting system CALIOPE (Baldasano et al., 2011; http://www.bsc.es/caliope/?language=en) with the Research LINE source dispersion model (R-LINE; Snyder et al., 2013) and its evaluation over the city of Barcelona, Spain. In Barcelona, chronic $NO_2$ exceedances have been recorded since the year 2000, and according to the local Public Health Agency about 68% of citizens were exposed to $NO_2$ levels above

the annual air quality limit value in 2016 (ASPB, 2017). Barcelona has a very high vehicle density (approx. 5500 vehicles $km^{-2}$) and the majority of passenger cars are diesel (67%) (Barcelona City Council, 2017). Located in the north east of the Iberian Peninsula, Barcelona is surrounded by the Mediterranean sea, two rivers and a mountain range. Due to its coastal emplacement, during the warm season, transport and dispersion of air pollutants within the city are dominated by the breeze blowing in from the sea during daytime and from the land during nighttime. This pattern persists under the presence of high-

pressure systems accompanied by clear skies and warm temperatures in the summer season. In contrast, the winter season is dominated by north western advections typically cleaning the atmosphere of the city (Jorba et al., 2011). Our aim is to produce more accurate $NO_2$ concentrations with CALIOPE-Urban, the coupled system, than with the mesoscale system alone and give a more realistic representation of $NO_2$ spatial distribution and temporal variability across the city. To achieve these objectives a set of system enhancements have been implemented: an adaptation of R-LINE to dense urban areas (e.g. street

canyons); a background model to estimate over background roof-level concentrations; a parametrization of the vertical mixing to estimate background concentrations within the street that considers atmospheric stability and urban geometry; and a local surface roughness parametrization to estimate turbulent parameters within a street canyon. The mesoscale system has been executed using the operational forecast configuration. We compare the estimated temporal variability of $NO_2$ concentrations from the coupled modeling system with those derived from CALIOPE and with ambient street-level measurements (i.e. 5 traffic



site and 1 urban background site) in April and May 2013. Its spatial variability is evaluated using a two-week measurement campaign that covered Barcelona with 182 NO$_2$ passive dosimeters for two weeks in February and March 2017.

## 2 Methods

CALIOPE-Urban estimates hourly NO$_2$ concentrations by coupling the CALIOPE mesoscale air quality forecasting system,
5 providing background concentrations, meteorological data and road-link traffic emissions, with the R-LINE dispersion model adapted to street canyons. Here we introduce and describe the components of the coupled model as depicted in Fig. 1.

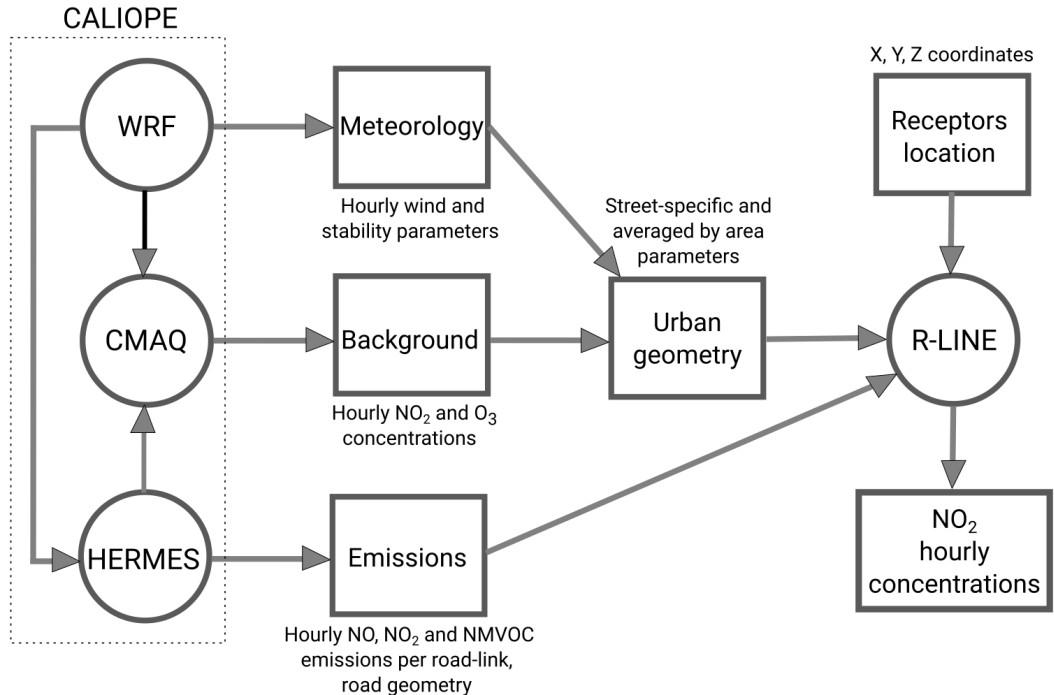

**Figure 1.** CALIOPE-Urban workflow. Models are represented by circles and data by rectangular shapes. CALIOPE is left untouched. Meteorology and background from WRF and CMAQ are combined with urban geometry to create inputs for R-LINE. R-LINE dispersion is left untouched, after adjusting meteorology and surface roughness for local urban geometry.

## 2.1 Mesoscale air quality forecasting system-CALIOPE

CALIOPE (Baldasano et al., 2011) integrates the Weather Research and Forecasting model version 3 (WRF; Skamarock and Klemp, 2008), the High-Elective Resolution Modelling Emission System (HERMESv2.0; Guevara et al., 2013), the Commu-
10 nity Multiscale Air Quality Modeling System version 5.0.2 (CMAQ; Byun and Schere, 2006) and the mineral Dust REgional Atmospheric Model (BSC-DREAM8b; Basart et al., 2012). The mesoscale system is run over Europe at a 12 km × 12 km



horizontal resolution, Iberian Peninsula at 4 km × 4 km, and the Catalonian domain, including Barcelona, at 1 km × 1 km. CALIOPE results have been evaluated in detail elsewhere (e.g., Pay et al., 2014).

In our system, WRF uses the Global Forecasting System (GFS) model initial/boundary conditions from the National Centers for Environmental Prediction (NCEP) to forecast the mesoscale meteorological conditions. Three nested domains are designed

to provide a final high-resolution run over Catalonian domain. In the vertical, WRF is configured with 38 sigma layers up to 50 hPa, where 11 cover the planetary boundary layer (PBL). Our WRF setup utilizes the Rapid Radiation Transfer model for long-wave radiation and Dudhia for short-wave, the Kain Fritsch cumulus parameterization, the single-moment 3-class microphysics scheme, the Yonsei University PBL scheme, and the Noah land-surface model based on the CORINE land-use data from the year 2006.

For the mesoscale model, pollutant emissions are obtained from the high resolution emission model HERMESv2.0 gridded up to 1 km × 1 km and temporal (1h) resolution. HERMESv2.0 estimates atmospheric emissions for Europe and Spain according to the Selected Nomenclature for Air Pollution (SNAP) and taking the year 2009 as the reference period. Emissions are estimated for nitrogen oxides ($NO_X$), non-methane volatile organic compounds (NMVOCs), sulphur dioxide, carbon monoxide, ammonia, total suspended particles, PM10 and PM2.5 fractions. The final model output consists of hourly, gridded and

speciated emissions according to the CB05 chemical mechanism used by the chemical transport model CMAQ. For Europe, HERMESv2.0 implements a SNAP sector-dependent spatial, temporal and speciation treatment of the original annual EMEP gridded emissions (Ferreira et al., 2013). For Spain, the model uses a bottom-up approach for pollutant sources including point (e.g. power plants, industries), maritime (e.g. ports), air traffic (e.g. airports), agricultural machinery (e.g. tractors and harvesters) and road transport. For the rest of pollutant sources a combination of top-down approaches (i.e. residential/commercial

combustion; energy consumption statistics combined with a population map) and downscaling methodologies (i.e. use of solvents, extraction and distribution of fossil fuels; specific spatial proxies and temporal profiles assigned to the Spanish National Emission Inventory by categories at third level of SNAP) is adopted. The results of the HERMESv2.0 model have been used to support several air quality evaluation and planning studies (e.g., Baldasano et al., 2014; Soret et al., 2014) as well as emission inventory intercomparison exercises (Guevara et al., 2017).

The chemical transport model used in the CALIOPE system is the CMAQv5.0.2. It uses the CB05 gas-phase chemical mechanism, the AERO5 aerosol scheme, and an in-line photolysis calculation. CMAQ vertical levels are collapsed from the 38 WRF levels to 15 layers up to 50 hPa with six layers falling within the PBL. We use as boundary conditions for the European domain MOZART-4.

## 2.2 Street scale dispersion model: R-LINE

R-LINE is a near-road Gaussian dispersion model (Snyder et al., 2013) that incorporates state-of-the-art Gaussian dispersion curves (Venkatram et al., 2013) to simulate dispersion of road source emissions. The model resolves either numerically or analytically the integration of the contributions of point sources along a street segment (Snyder and Heist, 2013). The first option is more accurate and the latter spends less time computing dispersion. The analytical version is best suited for near-ground level sources and receptors. In order to estimate $NO_2$ concentrations R-LINE incorporates a chemistry module to

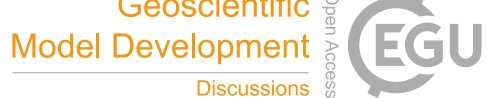



resolve simple NO to $NO_2$ chemistry with the Generic Reaction Set (GRS; Valencia et al., 2018). R-LINE has been applied to estimate exposure to traffic-related air pollutants in a large scale study in Detroit, United States (Isakov et al., 2014). However, to our knowledge it has not been applied to European cities, where street canyon morphology dominates. Hence, in order to apply R-LINE over Barcelona its meteorology has been adapted to street canyons as described in Sect. 2.3.1 and the background

concentrations are obtained from CMAQ model considering local meteorology and urban geometry as described in Sect. 2.3.3.

## 2.3  Coupling CALIOPE with R-LINE

CALIOPE and R-LINE are coupled offline, first CALIOPE is run over Europe, Iberian Peninsula and Catalonia and then R-LINE is executed for Barcelona city. This approach presents two main challenges that have already been highlighted in the research literature: (1) downscaling regional meteorology to street scale to drive pollutant dispersion; and (2) obtaining

background concentrations from the mesoscale model without double counting traffic emissions in regional and street scale models. In addition to these challenges, we consider relevant to couple meteorology and background concentrations in a consistent way, taking into account atmospheric stability and urban geometry when estimating background contribution within urban streets. Here we describe our methodology when coupling the models to mitigate these challenges.

### 2.3.1  Meteorology

WRF bottom layer results are assumed to represent over roof wind and stability conditions because its mid-point height (20.3 m) is similar to average building height ($\overline{bh}$) in a typical neighbourhood of Barcelona (e.g. Eixample district; 20.7 m). WRF is executed consistently with the forecasting air quality system CALIOPE, giving a constant surface roughness ($z_0$) equal to 1 m over the urban area. In order to apply R-LINE over Barcelona, its meteorology has been adapted to street canyons. We have developed a methodology to estimate specific $z_0$ based on urban geometry (e.g. building height, street width). Once $z_0$

is adjusted, the displacement height ($dispht$), friction velocity ($u*$), convective velocity scale ($w*$), PBL height, and Monin-Obukov length ($L$) are re-calculated (Cimorelli et al., 2005). The increase in $z_0$ generally leads to a larger $dispht$, $u*$, $w*$, and PBL height. Therefore, $L$ is less stable and atmospheric conditions are more convective. Ultimately, these adjustments have an effect on the way the winds are profiled and on the rate of dispersion of the roadway emissions within the urban area.

The geometrical parameters used for $z_0$ calculation are divided into two categories: (1) averaged over an area of 250m ×

250m (planar building density, $bd$; average building height, $\overline{bh}$; and building height standard deviation, $bhdev$); and (2) specific aspect ratio ($a_r$) for each street segment consisting of street-averaged building height divided by street width. The geometrical parameters are calculated from a Barcelona City Council dataset containing 2-D geometries and number of floors for each building (Barcelona City Council, 2016), assuming 3 m height for each floor.

To estimate specific $z_0$ for each street segment we propose a new morphometric method inspired by previous studies in the

literature. $z_0$ is composed by the WRF's background roughness ($z_{0bg}$) and the one estimated locally (Eq. 1), which incorporates building height influence through the $range$ parameter scaled by two parabolic ratios based on aspect ratio ($a_{rr}$) and building density ($bd_r$). The $range$ parameter (Eq. 2) and $z_0$ increase with $\overline{bh}$ following most morphometric methods (e.g. Macdonald et al., 1998). In addition, $range$ and $z_0$ increase with an increasing $bhdev$. This assumption is based on Kent et al. (2017),





who compared nine methods to estimate $z_0$ concluding that methods considering height variability through $bhdev$ (i.e. a higher $bhdev$ brings an increase of $z_0$) provide better results (e.g. Kanda et al., 2013). The parameter $C$ multiplying the equation for $range$ calculation is an empirical constant set to 1/20 after calibrating the system with the $NO_2$ measurements used in this work for CALIOPE-Urban evaluation. $dispht$ is calculated following R-LINE methodology given a factor of displacement

height ($facdispht$) equal to 5 (Eq. 3) as suggested by Snyder and Heist (2013).

$$z_0 = a_{rr} \cdot bd_r \cdot range + z_{0bg} \tag{1}$$
$$range = C \cdot (\overline{bh} + bhdev) \tag{2}$$
$$dispht = facdispht \cdot z_0 \tag{3}$$

To model the influence of building density and aspect ratio, we use Oke (1988) finding based on wind tunnel and experimen-

tal studies. Oke concluded that over-roof air roughness and satisfactory dispersion within the street canyon are maximum under similar geometrical conditions. Specifically, showing that an $a_r$ equals 0.65 and a $bd$ equals 0.25 give maximum roughness for overlying air and optimal dispersion conditions in the street canyon.

In practice, $z_0$ increases with an increasing $a_r$ to a maximum of $a_r = 0.65$ and decreases for $a_r > 0.65$ (Eq. 4). Additionally, an increasing $bd$ produces higher $z_0$ until a maximum at $bd = 0.25$ and decreases for higher $bd$ (Eq. 5). We model these ratios

using parabolic shapes ranging from 0 to 1. Both urban characteristics are modelled using one parabola to the left of the maximum and another to the right due to the non symmetrical distribution of the parameter values within Barcelona city (see Fig. A1 in Appendix A). The parabolic ratios will be maximum (i.e. equal to 1) if the roughness effect is maximum. The ratios are prevented from having negative values by setting a minimum of 0.

$$a_{rr} = \begin{cases} 1.0 - 2.3 \cdot (a_r - 0.65)^2 & \text{if } ar \text{ is} <= 0.65 \\ max(0, 1.0 - 1.38 \cdot (a_r - 0.65)^2) & \text{if } ar \text{ is} > 0.65 \end{cases} \tag{4}$$

$$bd_r = \begin{cases} 1.0 - 16.0 \cdot (bd - 0.25)^2 & \text{if } bd \text{ is} <= 0.25 \\ max(0, 1.0 - 8.1 \cdot (bd - 0.25)^2) & \text{if } bd \text{ is} > 0.25 \end{cases} \tag{5}$$

In addition to the $z_0$ adjustment, we adjust the wind speed and direction to represent more closely the winds blowing down the street as constrained by the buildings, which is called "channelling" (similarly to Fisher et al., 2005). We have adapted R-LINE to incorporate the orientation of roadways (and thus the buildings) where the wind direction follows the street direction. This leads to a recalculation of the wind direction and speed for each roadway before emissions are dispersed within a city.

Wind speed channelling is parametrized following Soulhac et al. (2008) who showed that mean velocity along a canyon for



any wind direction is directly proportional to the cosine of the angle between street direction and over roof wind direction (i.e. angle of incidence).

$$ws_{ch} = ws_{bh} \cdot max(0.1, abs(cos(\theta))) \tag{6}$$

where $ws_{ch}$ means channelled wind speed at roof level, the wind speed at roof level ($ws_{bh}$) is taken from the WRF bottom layer

in m/s and $\theta$ is the angle of incidence. The minimum value of the right component is set to avoid an unrealistic zero value for wind speed. Its value of 0.1 is defined in line with Kastner-Klein et al. (2001), who showed that minimum longitudinal mean flow velocity component at canyon top is equivalent to 0.12 times the above canyon wind speed for perpendicular over roof winds according to their wind tunnel experiments. Then, to estimate wind speed at street level a logarithmic profile incorporated within R-LINE that is based on similarity theory (Monin and Obukhov, 1954) is used.

**2.3.2  Emissions**

HERMESv2.0 provides hourly $NO_X$ and NMVOCs road transport emissions at the road link level, which are used by the R-LINE model algorithms to account for $NO_2$ near-road chemistry (Valencia et al., 2018). Road transport emissions (i.e. exhaust, evaporative, wear and resuspension) are estimated combining the Tier 3 method described in the EMEP/EEA air pollutant emission inventory guidebook (fully incorporated in version 5.1 of the COPERT IV software) with a digitized traffic network

that contains specific information by road stretch for daily average traffic, mean speed circulation, temporal profiles and vehicular park profiles. We note that HERMESv2.0 uses COPERT IV, which does not incorporate revised emission factors of $NO_X$ related to diesel gate. Hence, $NO_X$ emissions from traffic may be underestimated. Input activity data is obtained by combining different datasets, including traffic data from the Barcelona Automatic Traffic Counting Equipment and vehicle composition profiles derived from a Remote Sensing Campaign performed in different areas of Barcelona during 2010 (Barcelona City

Council, 2010). In Barcelona, higher levels of traffic emissions are found in the city center and in the highways surrounding the city (Fig. 2). In order to produce emissions in $gm^{-1}s^{-1}$ for straight street segments as required by R-LINE, we converted the digitized road network curved segments in HERMES to straight segments with no intermediate vertices using the Douglas-Peucker algorithm in the QGIS simplify geometries tool (QGIS Development Team, 2017).





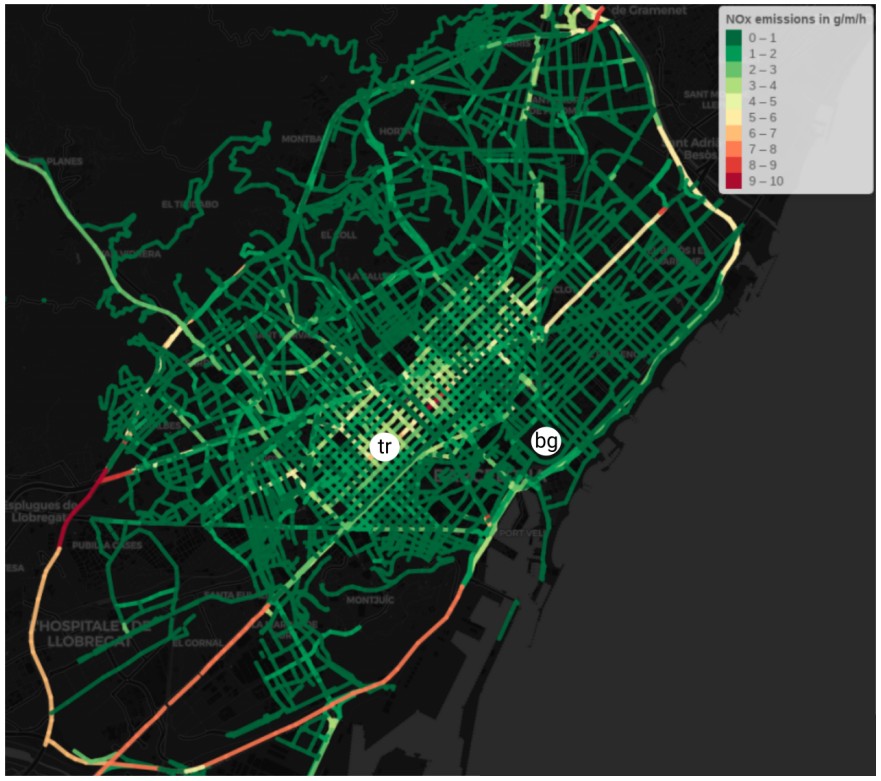

**Figure 2.** $NO_X$ emissions in $gm^{-1}h^{-1}$ in Barcelona city at 7 UTC on 11/4/2013 and location of the two fixed monitoring stations used to estimate $NO_2/NO_X$ ratio. White circles with letters inside represent the stations: $tr$ is Eixample traffic station and $bg$ is Ciutadella Park urban background station.

We have estimated $NO_2/NO_X$ ratio following Carslaw and Beevers (2004), which produces an approximation to the $NO_2$ primary contribution. This method relates total $O_X$ ($NO_2 + O_3$) to total $NO_X$ ($NO_2 + NO$) in a traffic monitoring station subtracting $O_X$ and $NO_X$ from a background site in order to remove the effect of background and to only calculate the contribution at the traffic site. As the traffic station we used Eixample site and as the urban background station Ciutadella Park (see Fig. 2), which is located upwind of the dominant wind direction. Figure 3 compares $O_X$ to $NO_X$ in Eixample after subtracting the background represented by Ciutadella from the beginning of October to end of February for years 2012 to 2016. The photochemical season (April-September) is not used to avoid greater scatter than it is found in the winter months as shown by Clapp and Jenkin (2001). The $O_X$ slope value of 18.9% is considered an estimate of the potential primary $NO_2$ contribution from vehicles on Eixample traffic station. This value is consistent with studies conducted in other cities with high diesel vehicle fleet (e.g., Carslaw et al., 2016; Wild et al., 2017) and is assumed to represent the $NO_2/NO_X$ ratio in Barcelona in the present work.





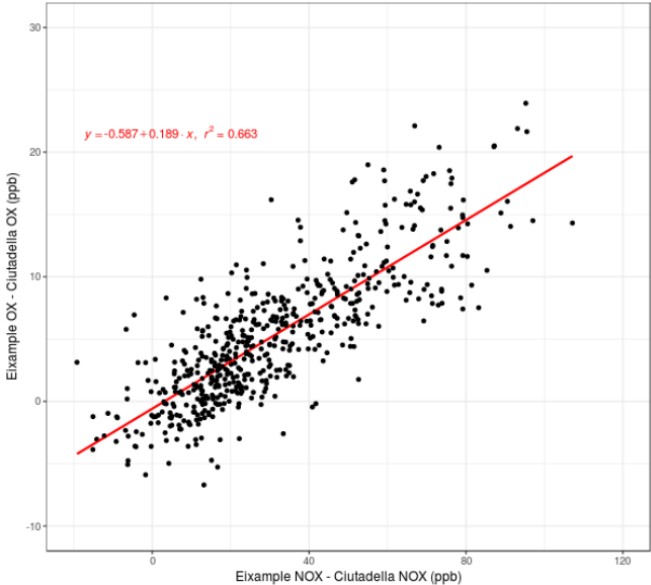

**Figure 3.** Scatter plot showing daylight mean $O_X$ and $NO_X$ relation of the difference between Eixample and Ciutadella stations from begin of October to end of February for years 2012 to 2016.

### 2.3.3 Background concentrations

We use the upwind Urban Background Scheme (UBS) to avoid the double counting of traffic emissions when coupling the mesoscale with the street scale model. The UBS makes a selective choice of CMAQ cells as sketched in Fig. 4 to estimate over roof background concentrations. For each hour, a polygon covering upwind air masses (white) is created. In the figure, the average distance traversed by air masses during an hour (10.8 km) is estimated for WRF's bottom layer wind speed (3 m/s in the image). Squares falling within the scheme polygon represent CMAQ cells and their color refer to cell pollutant values (e.g. $NO_2$ at peak traffic hours may be higher within the city than over the Mediterranean sea). Grid cell values falling over the scheme polygon are inverse-distance averaged to produce the background estimate of the scheme. Under calm conditions, only the upwind cell is chosen. This method is inspired by Berkowicz (2000) who apply a similar concept based on air masses trajectory to develop a background model.





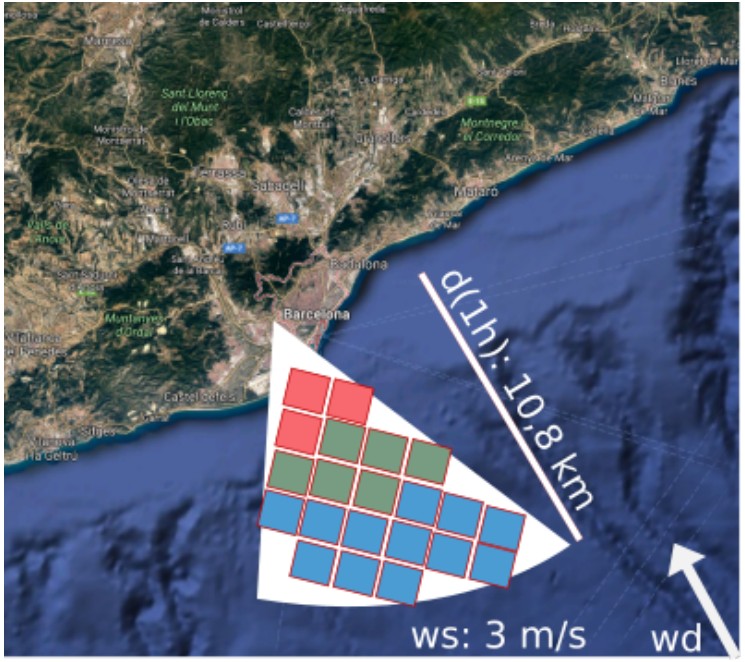

**Figure 4.** Upwind urban background scheme concept.

Background concentrations are required at each receptor in CALIOPE-Urban. Urban dispersion models are typically run at a very high spatial resolution (e.g. 20 m x 20 m). Running the UBS every 20 meters would have a high computational cost due to its spatial computations and background concentration values are not expected to vary substantially over tens of meters because CMAQ produces results with 1 km × 1 km spatial resolution. Hence, we first run the UBS to produce background concentration values at CMAQ grid cell centroids, then we apply a bilinear interpolation method to provide background at very high spatial resolution.

In addition to the UBS we implement a background decay method to calculate the surface level background concentrations assuming that the UBS provides the concentration at rooftop level. The relationship between rooftop and surface level concentrations is assumed to depend on atmospheric stability, localized surface roughness and urban geometry. The ratio of wind speeds at surface and rooftop levels ($ws_{sfc}/ws_{bh}$) estimated by R-LINE using similarity theory (Monin and Obukhov, 1954) is used as a proxy for the vertical mixing. Using this ratio, we calculate $fac_{bg}$ that represents the adimensional vertical mixing variable that is multiplied to rooftop background concentration to obtain surface level background concentration at a given height. In order to diminish the effect of afternoon underestimations from the regional system near traffic, background levels under convective situations are enhanced. We consider the upward heat flux at the surface ($hflux$) as representing convective conditions for values higher than 0.30. This value is set to exclude slightly stable night hours with low positive $hflux$



values mainly caused by the urban heat island (i.e. Barcelona city has been found to be 2.9 °C warmer than its periphery by Moreno-Garcia, 1994). The following parametrization is used for cases with $bd$ higher than 0.1,

$$fac_{bg} = \begin{cases} 1 - F + ws_{sfc}/ws_{bh} \cdot F & \text{if } hflux \text{ is} > 0.30 \\ ws_{sfc}/ws_{bh} & \text{if } hflux \text{ is} <= 0.30 \end{cases} \tag{7}$$

where $F = m + abs(0.25 - bd)$, being $m$ an empirical parameter set to 0.35 after system calibration with $NO_2$ measurements;
$hflux$ is upward heat flux at the surface (W m$^{-2}$). Surface background concentrations for convective situations are maximum for $bd$ equal 0.25 consistently with $z_0$ estimation in Sect. 2.3.1. On the other hand, we assume that for $bd$ close to zero, surface background concentrations tend linearly to rooftop level background concentrations. This linear transition starts when $bd$ equals 0.1 and ends when the surface background gets over roof value for $bd$ equals 0. The threshold $bd = 0.1$ is based on Grimmond and Oke (1999), who set it as an inferior limit for real cities and show that below this value an isolated flow regime governs.
Within this regime, street level and over roof air is well mixed due to the low building density. Hence, for cases with $bd$ equal or lower than 0.1, $fac_{bg}$ tends linearly to 1 following,

$$fac_{bg} = \begin{cases} 1 - 5 \cdot bd + ws_{sfc}/ws_{bh} \cdot (5 \cdot bd) & \text{if } hflux \text{ is} > 0.30 \\ 1 - 10 \cdot bd + ws_{sfc}/ws_{bh} \cdot (10 \cdot bd) & \text{if } hflux \text{ is} <= 0.30 \end{cases} \tag{8}$$

Equations 8 are linear variations between the point at $bd = 0$ and $fac_{bg} = 1$, and the point at $bd = 0.1$ with the corresponding $fac_{bg}$ value from the Eq. 7.

## 2.4 Execution setup

We have run CALIOPE-Urban for receptors as far as 250 metres from roads with sufficient Annual Average Daily Traffic (AADT) (i.e. 2000 vehicles/day following Jensen et al. (2017)) and receptors further away receive directly CMAQ values interpolated. The 250 m limit is chosen as similar but less restrictive (i.e. to allow longer distances under stable hours) than the one used in Beevers et al. (2012) who used 225 m for London. To smooth out the variation between system outputs, we define
a transition area (i.e. 140 m to 250 m) where receptors are given concentration values weighted by distance. For temporal and spatial evaluation runs, we locate receptors at the specific coordinates of the measurement sites.

To obtain high resolution concentration maps for the entire city, we set the spatial context as the minimum rectangle where Barcelona municipality is contained and extended it by 250 m buffers that include the highways surrounding the city. The context is covered by a regular receptor grid of 10 meter resolution. R-LINE execution loops over each hour, road and receptor
to estimate the contribution from each source to each receptor.

Aiming to understand the impact on accuracy of the local parametrization for background and meteorology and the impact of using the analytical approach for dispersion, we have run CALIOPE-Urban with different configurations. In Table 1, we describe the different scenarios that have been run. As seen in the table, the CALIOPE-Urban and the CALIOPE-Urban





Analytical configurations make use of the developed local parametrizations for background and meteorology. In contrast, the CALIOPE-Urban-nl (Non Local) configuration does not apply the local parametrizations for background and meteorology. Instead, it uses as background the UBS output without vertical mixing and it omits the use of wind channelling and specific stability parameters for each street segment based on local $z_0$. We show this configuration's results in order to understand if the new implementations in this work contribute substantially to improve the system's ability to simulate $NO_2$ concentrations in Barcelona. R-LINE dispersion algorithm options (i.e. analytical and numerical) are described in Sect. 2.2. For meteorological options, we refer to Sect. 2.3.1. The background method is described in Sect. 2.3.3.

**Table 1.** Description of the execution setup. Execution time is for the entire city of Barcelona during one hour running CALIOPE-Urban (i.e. only the urban system, after CALIOPE run completion) over 11251 street segments and 965458 receptors at 10 m $\times$ 10 m spatial resolution.

| Configuration Name | Dispersion algorithm | Meteorology | Background | Execution time |
|---|---|---|---|---|
| CALIOPE-Urban | Numerical | Local | Local | 88 minutes |
| CALIOPE-Urban-nl | Numerical | Non Local | Non Local | 56 minutes |
| CALIOPE-Urban Analytical | Analytical | Local | Local | 44 minutes |

## 3 Observational datasets

We use three datasets of observations to evaluate the performance of CALIOPE-Urban to reproduce the temporal and spatial variation of $NO_2$ concentrations within Barcelona city. Fig. 5 shows the locations of measurements used in this study, which are described below.





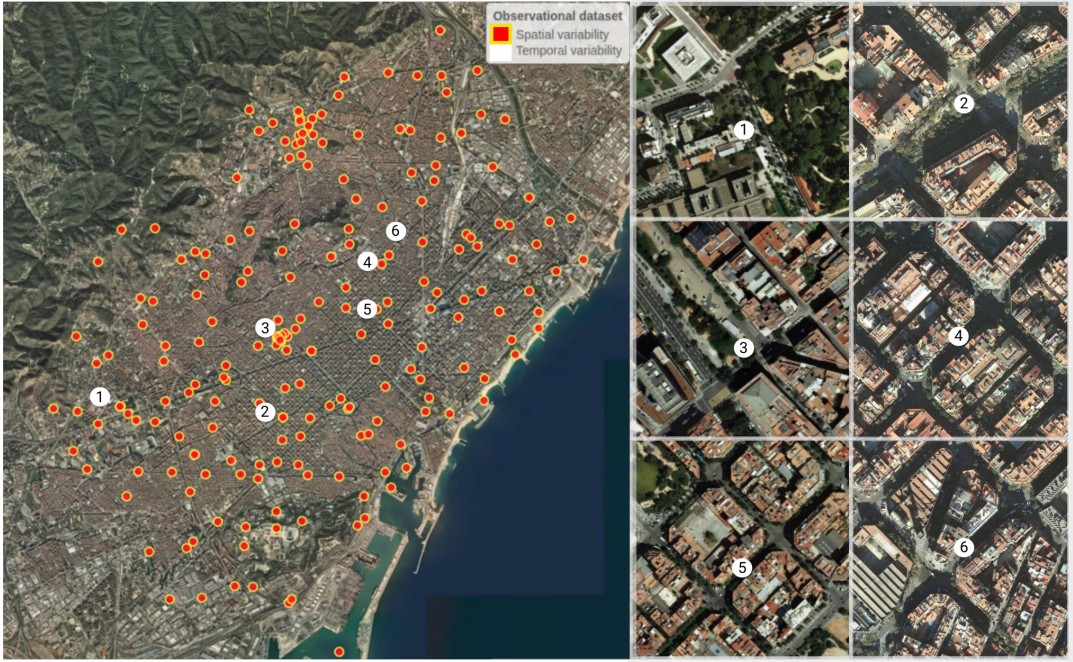

**Figure 5.** Passive dosimeters and monitoring sites location used in the evaluation of CALIOPE-Urban in this work. Red dots with yellow border represent passive dosimeters (spatial performance) location and white numbered dots depict monitoring site emplacements (temporal variability). White dots numbered 1 (Palau Reial), 2 (Eixample) and 3 (Gràcia-Sant Gervasi) are air quality monitoring sites and 4 (213 Industria Street), 5 (445 Valencia Street) and 6 (309 Industria Street) correspond to mobile units.

### 3.1 NO$_2$ temporal variability: Street canyon campaign and permanent XVPCA network

To evaluate the NO$_2$ temporal variability we use hourly NO$_2$ concentrations reported by the official monitoring network in Catalunya (XVPCA) and from an experimental campaign conducted using mobile units in April and May 2013 in Barcelona (Amato et al., 2014). The official monitoring network has 10 stations in Barcelona and only two of them (i.e. Gràcia-Sant

5   Gervasi and Eixample) are considered representative of near traffic conditions and provide NO$_2$ hourly levels. Measured data from 3 sites of the official network are used in this study: Eixample and Gràcia-Sant Gervasi (traffic) and Palau Reial (background). Both traffic sites are located in complex wide areas where several streets intersect (see sites 2 and 3 in Fig. 5 and in description Table 2). Palau Reial station (i.e. site 1 in Fig. 5) is located in a medium $bd$ area of the city, 300 metres away from a heavily trafficked street. This dataset is complemented with observations from a experimental campaign where mobile units

10  placed at the parking lane of several street segments measured air quality parameters at 3 m height. For this study, we used data gathered every 30 minutes and aggregated to hourly levels for homogeneity at 213 Industria Street, 309 Industria Street and 445 Valencia Street. These streets present a marked canyon pattern (see sites 4, 5 and 6 in Fig. 5 and description table) where aspect ratio is approximately 1. In Barcelona, different street geometrical patterns cohabit. For example, the Eixample district, which has the highest number of inhabitants and the greatest population density (33000 inhabitants km$^{-2}$), is characterized by



a marked street canyon pattern. Most of its canyons are about 20 to 25 m high and 20 m wide (i.e. $a_r$=1 and higher than 1). Experimental campaign sites are considered traffic sites in this work because they are exposed to similar AADT and vehicles km$^{-2}$ compared to official traffic sites as shown in the table below. We apply Eq. (9) to obtain vehicles km$^{-2}$, a variable that describes traffic density in an area of 1 km$^2$.

$$vehicles \cdot km^{-2} = \sum_{n=1}^{st} Vehicles/s \cdot length \tag{9}$$

To obtain the amount of vehicles per second, AADT is divided by 3600 * 24 and multiplied by a temporal factor (i.e. 1.47) representing a typical factor for morning traffic peak in Barcelona. $Length$ is street length in metres. $st$ is the number of streets over the circular area of 1 km$^2$ centered in the measurement site.

**Table 2.** Morphometric and traffic description of measurement sites used in CALIOPE-Urban evaluation. AADT from the nearest street is considered. Vehicles km$^{-2}$ estimated following Eq. (9). Palau Reial vehicles km$^{-2}$ is not included because it is an urban background site not directly exposed to high traffic.

| Site | $a_r$ | $\overline{bh}$ | $bd$ | $bhdev$ | $z_0$ | AADT | vehicles km$^{-2}$ |
|---|---|---|---|---|---|---|---|
| 1. Palau Reial | 0.12 | 14.6 | 0.12 | 6.4 | 1.27 | 3900 | - |
| 2. Eixample | 0.00 | 21.1 | 0.40 | 8.4 | 1.03 | 41000 | 5666 |
| 3. Gràcia-Sant Gervasi | 0.38 | 17.2 | 0.45 | 7.1 | 1.68 | 12700 | 3884 |
| 4. 213 Industria Street | 1.00 | 18.1 | 0.38 | 8.3 | 1.94 | 15200 | 3003 |
| 5. 455 Valencia Street | 0.86 | 19.5 | 0.32 | 7.2 | 2.20 | 32500 | 5978 |
| 6. 309 Industria Street | 1.03 | 17.0 | 0.31 | 8.1 | 1.97 | 12900 | 3320 |

## 3.2 NO$_2$ spatial variability: Passive dosimeters campaign across the city

With the objective of representing the NO$_2$ spatial variability, 212 passive dosimeters were located in Barcelona from 28/2/2017 to 15/3/2017 as depicted by red dots with yellow border in Fig. 5. In every km$^2$ there were at least two dosimeters, representing the background and traffic conditions, respectively, at 2.2-2.5 m height. The 100 background dosimeters were placed more than 10 m away from the road and the 112 traffic dosimeters were located at less than 3 m away from the road and at least 25 m away from intersections. To ensure the equivalence of measurements to standard conditions, these were corrected through comparison with reference equipment from several sites of the XVPQA network. After a preliminary inspection of the location of the dosimeters, we discarded data from 30 dosimeters to avoid results that could not be interpreted for several reasons (e.g. dosimeter and simulated road at different heights; highway covered by a tunnel near dosimeter location that is not considered in the emission inventory; lack of emission sources near dosimeter).

## 4 Results and discussion

Section 4.1 presents the temporal variability of NO$_2$ concentrations estimated by CALIOPE and CALIOPE-Urban compared to observations at the six sites described in Sect. 3.1. Section 4.2 describes the results in terms of the spatial variation during the





two-week passive dosimeter campaign described in Sect. 3.2. Model performance is quantified using performance measures as described by Chang and Hanna (2004) and using assessment target plots (defined in the FAIRMODE initiative, Janssen et al., 2017). The performance statistics used here are the geometric mean bias (GeoMean), the fraction of model results within a factor of two of observations (FAC2), the geometric standard deviation (GeoSD), the correlation coefficient (R), the mean bias

5 (MB) and the root mean square error (RMSE). The mathematical expressions of these statistics can be found in the Appendix C.

## 4.1 Temporal variation of NO$_2$ concentrations within urban streets

The scatter plots of Fig. 6 compare CALIOPE and CALIOPE-Urban outputs with observations based on hourly, daily mean and maximum modelled concentrations in the six sites described in Sect. 3.1 for April and May 2013. In general, CALIOPE-

10 Urban shows a greater agreement for hourly, daily means and maximum concentrations but tends to underpredict daily peak concentrations in sites not exposed to very high traffic intensity (i.e. sites where urban background contribution predominates like Gràcia-Sant Gervasi). During the study period most of daily maxima (i.e. 56 %) occur at morning or evening traffic peak times (i.e. 6-7 or 18-20 UTC) when atmospheric conditions are typically stable and traffic intensity is high.





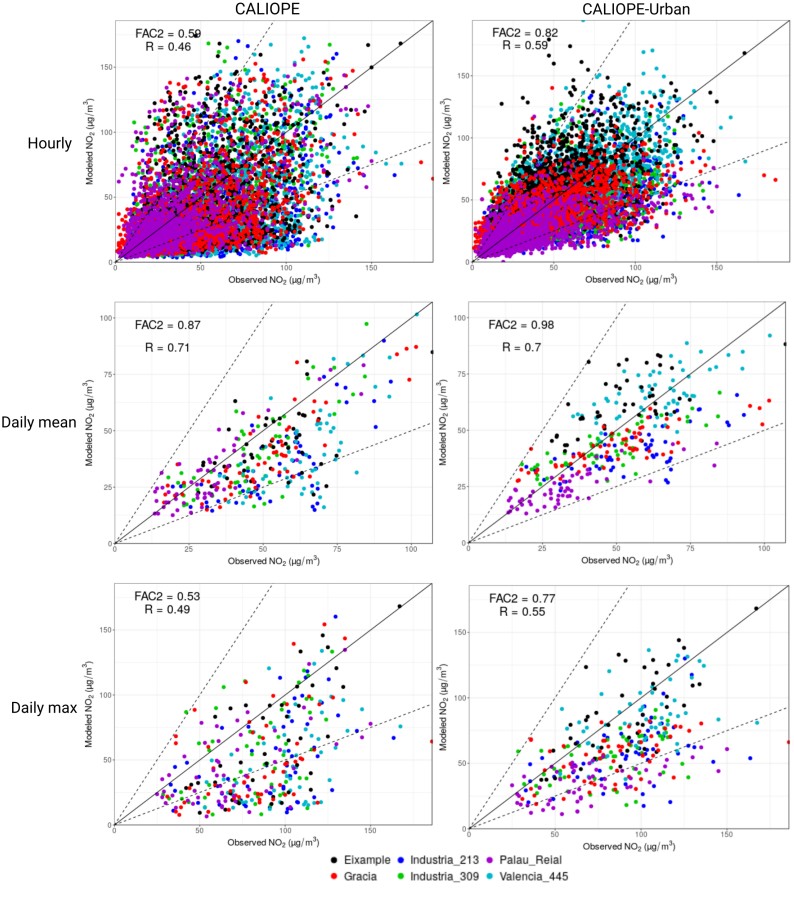

**Figure 6.** Scatter plot of hourly (top), daily mean (middle) and daily maximum (bottom) modelled concentrations against observed concentrations with colors representing monitoring sites for CALIOPE (left) and CALIOPE-Urban (right). Purple color represents Palau Reial, the urban background site. The other colors represent traffic sites as described in Sect. 3.1.

Table 3 shows the model performance statistics computed with hourly data, including CALIOPE-Urban-nl run. We compare CALIOPE-Urban and CALIOPE-Urban-nl to assess the difference in performance derived by the use of the local developments described in Sect. 2.3. All systems perform well at urban background sites and only CALIOPE-Urban gives good agreement with observations in traffic sites. The greatest difference between CALIOPE and CALIOPE-Urban systems performance is
5   produced at the 455 Valencia Street site due to its street canyon morphology ($a_r$ = 0.86). In this site, the mean transport is well resolved by the channelled winds, and its high AADT produces a high increase in traffic emissions within R-LINE. CALIOPE-Urban-nl largely overestimates $NO_2$ concentrations in this site for several reasons: it uses directly the output of UBS for background, instead of applying the vertical mixing that reduces background at street level specially under stable conditions; $z_0$ is given the WRF value ($z_0$ = 1.0), which is much lower than its locally estimated value (i.e. $z_0$ = 2.2, see Table




2) that enhances dispersion decreasing concentration levels; lastly pollutant dispersion is not channelled within the street, so higher contributions of nearby streets may be expected.

**Table 3.** NO$_2$ model evaluation statistics calculated at six sites (described in Sect. 3) for hourly concentrations during April and May 2013 for CALIOPE, CALIOPE-Urban and CALIOPE-Urban without local developments (CALIOPE-Urban-nl). Bold numbers represent model results with better performance for each statistic and site.

| Site | Method | FAC2 | MB | RMSE | GeoMean | GeoSD | r |
|---|---|---|---|---|---|---|---|
| 1. Palau Reial | CALIOPE | **0.73** | **-1.23** | 24.11 | **1.10** | 1.25 | 0.55 |
| | CALIOPE-Urban | 0.72 | -8.70 | **21.57** | 1.28 | **1.22** | **0.57** |
| | CALIOPE-Urban-nl | 0.67 | 3.61 | 26.34 | 1.28 | **1.22** | 0.55 |
| 2. Eixample | CALIOPE | 0.60 | **-8.57** | 35.14 | 1.35 | 1.35 | 0.34 |
| | CALIOPE-Urban | **0.86** | 9.38 | **26.70** | **0.83** | **1.11** | **0.55** |
| | CALIOPE-Urban-nl | 0.61 | 39.53 | 54.92 | 0.57 | 1.38 | 0.45 |
| 3. Gràcia-Sant Gervasi | CALIOPE | 0.55 | -10.95 | 31.95 | 1.38 | 1.39 | 0.47 |
| | CALIOPE-Urban | **0.79** | -7.39 | **25.11** | **1.07** | **1.19** | **0.52** |
| | CALIOPE-Urban-nl | 0.66 | **6.00** | 35.91 | 0.91 | 1.43 | 0.38 |
| 4. 213 Industria Street | CALIOPE | 0.52 | -19.13 | 35.13 | 1.79 | 1.54 | 0.44 |
| | CALIOPE-Urban | **0.78** | -13.62 | **26.55** | 1.30 | **1.17** | **0.57** |
| | CALIOPE-Urban-nl | 0.75 | **1.57** | 31.12 | **1.04** | 1.26 | 0.54 |
| 5. 455 Valencia Street | CALIOPE | 0.50 | -21.94 | 38.31 | 1.85 | 1.53 | 0.43 |
| | CALIOPE-Urban | **0.92** | **2.92** | **23.26** | **0.94** | **1.07** | **0.56** |
| | CALIOPE-Urban-nl | 0.79 | 23.72 | 42.29 | 0.75 | 1.19 | 0.47 |
| 6. 309 Industria Street | CALIOPE | 0.64 | -7.41 | 28.49 | 1.36 | 1.33 | 0.53 |
| | CALIOPE-Urban | **0.84** | **-4.60** | **22.72** | **1.05** | **1.13** | 0.53 |
| | CALIOPE-Urban-nl | 0.78 | 11.60 | 31.13 | 0.83 | 1.24 | **0.58** |

On the other hand, CALIOPE-Urban underestimations at 213 and 309 Industria Street and Gràcia-Sant Gervasi may be due to an unrealistically low AADT level on the street segment close to the site. We work with AADT data that is based on the outputs of the traffic model used by Barcelona City Council that may be underestimating traffic. Another explanation may be an underestimation of local background levels within the area mostly during the afternoon. The afternoon underestimations in the mesoscale system could be caused by an overestimation of the mixing that produces a too low background NO$_2$ concentration level. This issue is difficult to correct because background concentrations used in the system are dependent on mesoscale concentrations, which are underestimated during daytime. In Table B1 in the Appendix B, same statistics are computed for daily mean results, finding similar results as in the hourly analysis. In addition, the analytical version of CALIOPE-Urban is shown to produce similar results for hourly concentrations to the numerical version in Table B2 in the Appendix B. This result may be interesting for forecasting applications at urban scale that require high-resolution because the analytical dispersion algorithm spends approx. half the time computing in comparison to the numerical dispersion algorithm as shown in Table 1.

Figure 7 shows NO$_2$ assessment target plots for CALIOPE and CALIOPE-Urban. In the plots the centred root mean square error (CRMSE) for each measurement station is plotted against the normalized bias. Distance from circle origin gives an estimate for the model quality indicator (MQI; Thunis and Cuvelier, 2016) that measures general model accuracy depending on measurement uncertainty. MQI values below 1 (i.e. green shading area) are considered to comply with the model quality




objective. All sites in CALIOPE-Urban simulation fall within the green shading area (i.e. complying with FAIRMODE's model quality objective). In contrast, four out of six in CALIOPE lie within the green shading area clearly showing the positive effect of the street scale model in the coupled system.

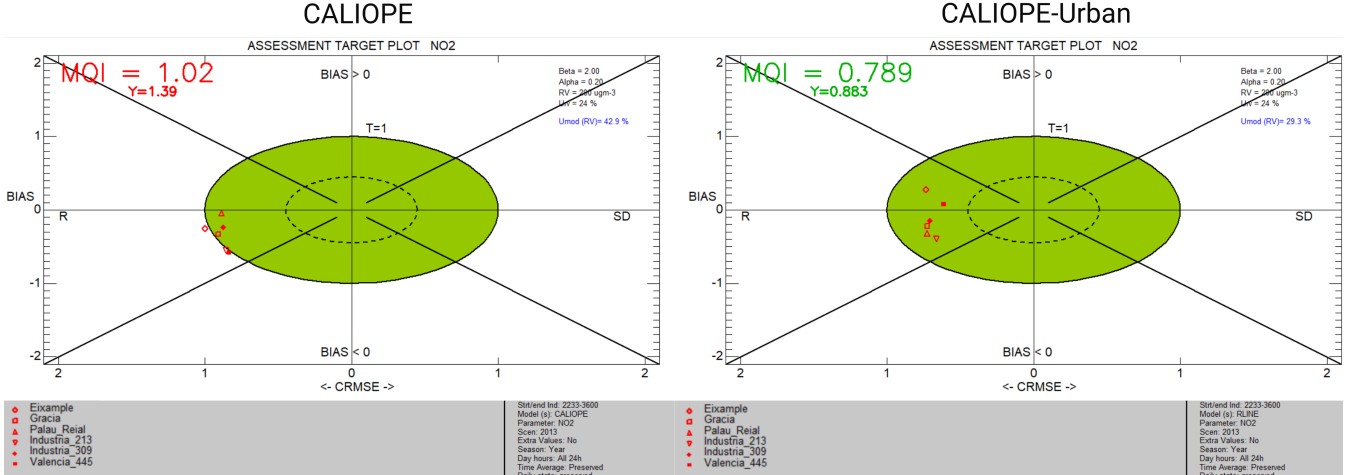

**Figure 7.** NO$_2$ model assessment target plots for CALIOPE (left) and CALIOPE-Urban (right). Symbols correspond to the six measurement sites described in Sect. 3.1 and the MQI for each site is represented by the distance between the circle origin and the site symbol.

Figure 8 shows averaged daily cycles for weekday and weekend periods for the six sites described in Sect. 3.1 for CALIOPE,

CALIOPE-Urban and CALIOPE-Urban-nl. In general, all systems show a significant change between weekday and weekend in accordance with observations. The overall dynamic is well reproduced by all systems but CALIOPE tends to underestimate the afternoon levels and to overestimate nighttime values. CALIOPE-Urban-nl overestimates nighttime values and morning peaks. CALIOPE-Urban partly corrects CALIOPE afternoon underestimations close to high traffic (i.e. Valencia Street and Eixample stations) but still underestimates in low traffic sites. CALIOPE's tendency to overestimate the evening peak and night values

may bring CALIOPE-Urban to generally overestimate on those hours as found in the literature near road sites (Hood et al., 2018). However, the vertical mixing implemented in CALIOPE-Urban decreases background concentrations mixing from aloft during night hours because under stable atmospheric conditions vertical mixing is reduced compared to daylight hours, which are more convective. This effect can be noticed in the difference between CALIOPE-Urban and CALIOPE-Urban-nl from 0 to 6 and from 18 to 23 (UTC) in traffic sites (i.e. sites 2,3,4,5 and 6 in Fig. 8), where CALIOPE-Urban concentration levels

correct the night overestimations seen in CALIOPE-Urban-nl. Such result shows the benefit of considering the vertical stability in the coupling procedure of the mesoscale and the street scale dispersion model.



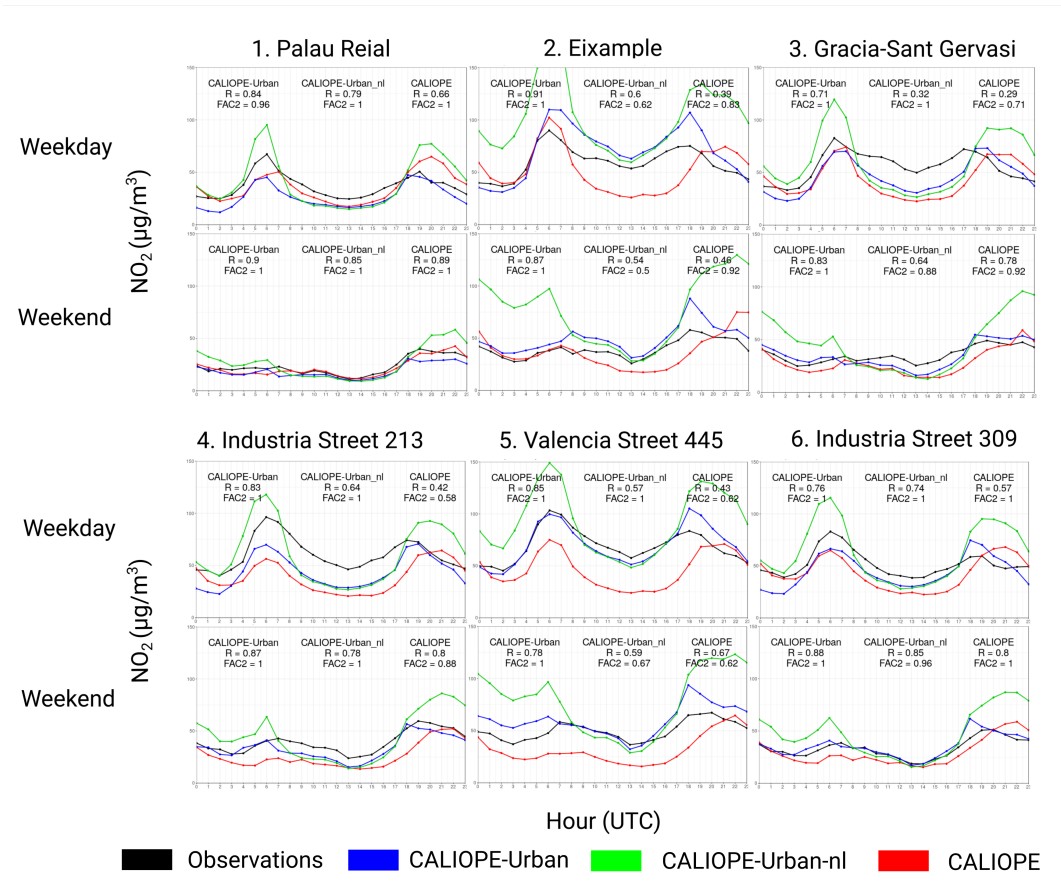

**Figure 8.** NO$_2$ average daily cycle at all sites described in Sect. 3.1 during April and May 2013 for weekday and weekend. Observations are represented in black coloured lines, red lines are CALIOPE, blue lines are CALIOPE-Urban and green lines represent CALIOPE-Urban without local developments (CALIOPE-Urban-nl).

There is a noticeable difference between CALIOPE-Urban's accuracy at 213 Industria Street and 445 Valencia Street given similar observations and CALIOPE levels at both sites. Although both sites are located in areas with considerable traffic activity, Valencia Street site has higher modelled traffic emissions, deriving in higher local pollutant concentrations, and a higher density of vehicles km$^{-2}$ as described in Table 2. Consequently, to improve CALIOPE-Urban accuracy an increase of local simulated traffic at 213 Industria Street site could bring a model accuracy improvement. However, the lack of observational traffic count data at the monitoring sites does not permit to explore the precision of the input AADT information considered in HERMESv2.0 at those locations.





## 4.2 Spatial variation of NO$_2$ concentrations across the city

We evaluate CALIOPE and CALIOPE-Urban NO$_2$ in terms of spatial variations across Barcelona city using measurements from 182 valid passive dosimeters as described in Sect. 3.2. Table 4 gives statistics at the 182 sites where passive dosimeters measured NO$_2$ concentrations for a two week period (28 February - 15 March 2017) for CALIOPE, CALIOPE-Urban and
CALIOPE-Urban-nl (without local developments).

**Table 4.** NO$_2$ model evaluation statistics calculated at 182 passive dosimeter valid sites (described in Sect. 3.2) during two weeks from 28 February to 15 March 2017 for CALIOPE, CALIOPE-Urban and CALIOPE-Urban-nl mean concentrations. Bold numbers represent model results with better performance for each statistic and site. Results are shown for all (including traffic and urban background), only urban background and only traffic sites.

| Sites | Method | FAC2 | MB | RMSE | GeoMean | GeoSD | r |
|---|---|---|---|---|---|---|---|
| All | CALIOPE | 0.92 | -14.00 | 21.88 | 1.30 | 1.08 | 0.36 |
|  | CALIOPE-Urban | 0.97 | **-7.26** | **17.21** | 1.20 | **1.06** | **0.70** |
|  | CALIOPE-Urban-nl | **0.98** | 15.24 | 28.30 | **0.81** | **1.06** | 0.69 |
| Background | CALIOPE | **1.00** | **-2.84** | **8.08** | **1.06** | **1.02** | 0.66 |
|  | CALIOPE-Urban | 0.97 | -7.34 | 12.71 | 1.23 | 1.06 | 0.54 |
|  | CALIOPE-Urban-nl | **1.00** | 10.09 | 15.03 | 0.82 | 1.04 | **0.72** |
| Traffic | CALIOPE | 0.81 | -25.57 | 30.74 | 1.60 | 1.63 | 0.22 |
|  | CALIOPE-Urban | **0.97** | **-7.17** | **20.62** | **1.17** | **1.06** | **0.53** |
|  | CALIOPE-Urban-nl | 0.96 | 20.17 | 36.76 | 0.81 | 1.08 | 0.50 |

Considering all sites, CALIOPE-Urban shows a much better correlation coefficient (0.70 vs 0.36) than CALIOPE due to its good performance at traffic sites. Compared to CALIOPE-Urban-nl their correlation is similar. If we consider only urban background sites, CALIOPE shows a greater correlation coefficient than CALIOPE-Urban (0.66 vs. 0.54) and a MB closer to 0. In addition, CALIOPE-Urban-nl gives a better correlation than both systems. A potential explanation for this result is related to
the error compensation shown in the temporal evaluation (Sect. 4.1). CALIOPE and CALIOPE-Urban-nl may compensate the underestimation during daytime with the overestimation during nighttime. In contrast, CALIOPE-Urban may not compensate the daytime underestimations with overestimated night values because the background is reduced due to low vertical mixing effect during nighttime (stable) hours. An enhanced daytime NO$_2$ background contribution would improve CALIOPE-Urban accuracy at urban background sites.

For traffic sites, CALIOPE shows a strong underestimation (MB = -25.57 µg m$^{-3}$) and CALIOPE-Urban gives MB levels closer to 0. CALIOPE-Urban underestimations may be influenced by afternoon underestimations and a misrepresentation of traffic emissions in some areas of the city. In contrast, CALIOPE-Urban-nl gives a high MB and the highest RMSE among the three systems. This tendency to over estimate near traffic of CALIOPE-Urban-nl may be due to the reasons stated in Sect. 4.1. In general, closer to intense traffic CALIOPE-Urban is very sensitive to emissions and its dispersion characterizes well
the spatial variability for the study period. Reproducing spatial gradients near intense traffic is crucial in a city like Barcelona given its high vehicle density and NO$_2$ concentration levels.



Figure 9 shows the difference between CALIOPE and CALIOPE-Urban results and measurements (top panels) and scatter plots at all sites (bottom panels) distinguished with colors by site type (e.g., traffic site, urban background site).

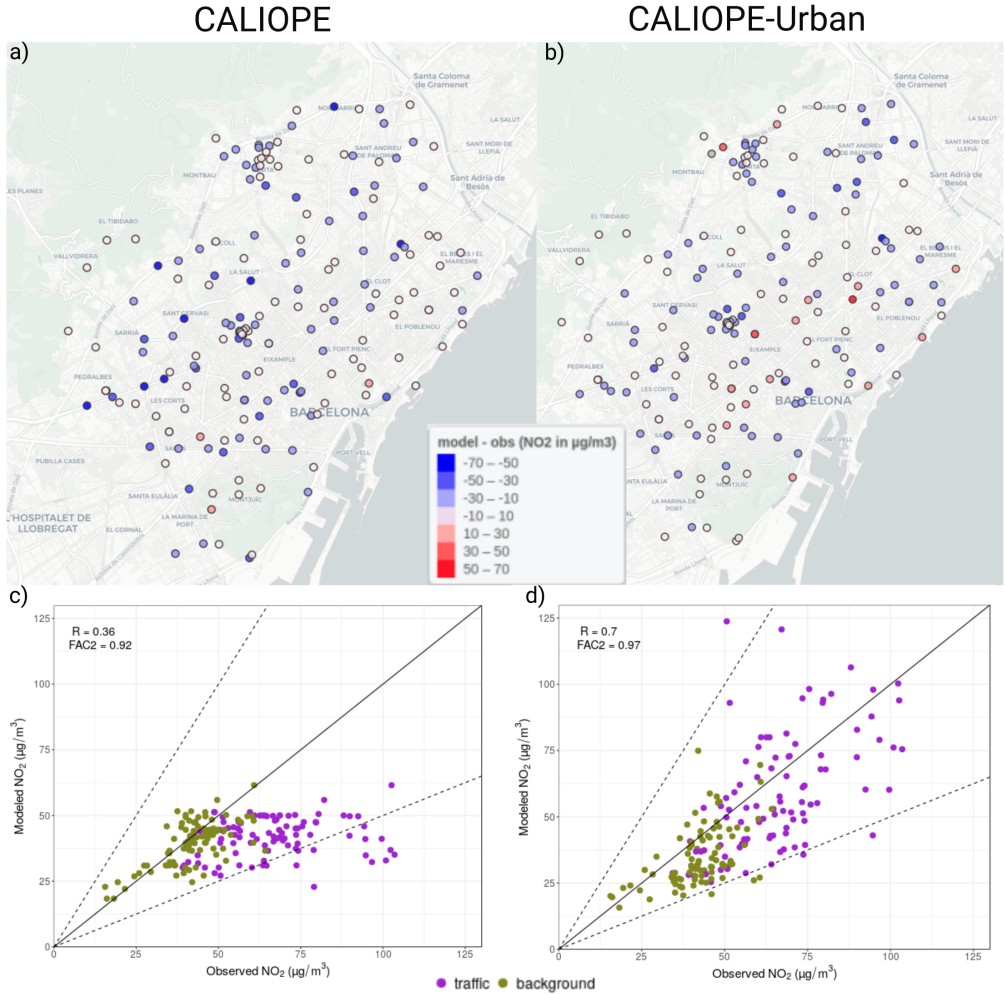

**Figure 9.** (Top) NO$_2$ concentrations difference (model - observations) for two week averaged values during dosimeters campaign in 2017, CALIOPE is left (a) and CALIOPE-Urban is right (b). (Bottom) scatter plot of modelled vs observed concentrations with colors representing site type (olive green for background and purple for traffic) for CALIOPE (c) and CALIOPE-Urban (d). Correlation (R) and agreement factor of 2 (FAC2) are computed for all sites.

In Fig. 9a the concentration difference map of CALIOPE shows an overall underestimation, represented by blue dots. This underestimation is found to be systematic in traffic sites in the scatter of Fig. 9c (purple dots), where modelled values barely exceed 50 µg m$^{-3}$ while most of the observed values at traffic sites are above that value. In contrast, the CALIOPE-Urban difference map (Fig. 9b) shows a more mixed picture with a broader representation of white dots (bias close to 0) but also more





red ones in the city center and close to the highways. For CALIOPE-Urban's scatter, most of the model results at traffic sites are within the 1:2 and 1:0.5 dashed lines, showing a better agreement at traffic sites than CALIOPE (Fig. 9d). In CALIOPE-Urban's difference map, we see a spatial pattern with average bias close to 0 in the city centre, where traffic is denser and close to the highways surrounding the city. The appearance of red dots may indicate that CALIOPE-Urban overestimates close to high trafficked areas while CALIOPE underestimate in these areas. This may be due to an overestimation of traffic emissions or background concentrations in these areas. In contrast, in locations where traffic is not very intense (see Fig. 2 for $NO_X$ emissions) CALIOPE-Urban shows systematic underestimations. This result may be derived from the systematic underestimation of midday $NO_2$ concentrations in low traffic areas as shown in Sect. 4.1.

### 4.3 Major uncertainty sources

Here we discuss potential sources of error in our model by analyzing episodes when the model was skillful compared with episodes when the model was not. Our analysis solely considers the meteorological and background concentration inputs as potential sources of error. While road traffic emission estimates may introduce large errors, we lack observations of traffic counts at the measurement site locations to properly assess them.

We calculated daily the RMSE of the hourly modelled $NO_2$ concentrations versus the observed values in the six sites described in Sect. 3.1 during the period April and May 2013. For each site we picked the ten days with highest RMSE as potential candidates and ten days with the lowest RMSE. We conducted this analysis for both CALIOPE and CALIOPE-Urban, finding that both systems share to a large extent the days with skill (4 out of 5 days) and without (3 out of 5). This result shows that the coupled system performance is highly dependent on the mesoscale model performance. To explore errors potentially caused by R-LINE inputs, in Fig. 9 we compare the five days with less skill (i.e. 11, 16, 17 April and 7,8 May) and the five days with more skill (i.e. 7, 20, April and 18, 19, 25 May) with observations for wind speed ($ws$), street level $NO_2$ and background $NO_2$.

On skillful days, winds are relatively strong and well represented in WRF (Fig. 10a). Poor skills appear when the observed wind speed is low. Because WRF largely underestimates wind speeds (Fig. 10b) and $NO_2$ concentrations are underestimated under calm conditions (Fig. 10d), other processes (e.g. atmospheric stability) may have a greater importance in this case. In our coupling under very stable atmospheric situations, dispersion is reduced and background injection from the overlying atmosphere is limited. This control mechanism adapts the system to specific street conditions, regulating dispersion and background injection. For these days, an extended observational dataset would be needed to better understand the model behaviour.



**Figure 10.** Boxplots by time of the day of good (left panels) and bad performance days (right panels) for CALIOPE-Urban inputs and observations with dots representing outliers. a) and b) represent WRF and observed wind speeds at Barcelona airport (10m height); c) and d) show observed and modelled $NO_2$ concentrations for the six sites in Sect. 3.1; e) and f) depict $NO_2$ observed concentrations at Ciutadella urban background station and background model averaged results at the six sites. Observed values are orange coloured and modelled results are blue. Light green represents background model results at surface level.





To analyze the background concentrations from the mesoscale simulation as a potential error source, we compared $NO_2$ observations from the Ciutadella urban background station with hourly modelled concentrations averaged over the six sites. The results shown in Fig. 10e,f represent concentrations provided by upwind CMAQ grid cells depending on wind speed and direction (blue) as described in Sect. 2.3.3 downscaled to surface level using the vertical decay method (green). As expected,

observed $NO_2$ concentrations on days with calm conditions and therefore poor skill are higher than on those with enhanced ventilation and better skills. The background model reproduces well the variation during both types of days but overestimates concentrations during nighttime (19-22 UTC), particularly during days with calm conditions. This problem is partially corrected by using the background vertical decay method as seen in Fig. 10f and in Fig. 8. In addition, $NO_2$ concentrations are underestimated at the beginning of the day (1-4 UTC). The fact that the averaged diurnal cycle in Fig. 8 shows similar error

patterns suggests that $NO_2$ background concentrations greatly influence $NO_2$ street level concentrations in agreement with Degraeuwe et al. (2017).

### 4.4    Hourly variation of street $NO_2$ concentrations

Hourly street $NO_2$ concentrations are expected to vary spatially and temporally with higher values close to intense traffic sites during rush hours. Figure 11 shows high resolution (10 m × 10 m) $NO_2$ concentration maps at four different hours on

Thursday 11th of April 2013 (i.e. 0, 7, 12 and 18 UTC). This day is chosen because it shows a marked diurnal cycle with maxima consistent with the morning and evening traffic peaks (i.e. 6-7 or 18-20 UTC). Higher concentrations are found at 7 and 18 UTC where high traffic emissions are concentrated (i.e. highways surrounding the city and city center) because traffic intensity is higher at these hours of the day and the atmosphere tends to be stable, making pollutant dispersion more difficult. On the other hand, lower concentrations are found at 0 UTC due to the lower traffic intensity and at 12 UTC. At 12 UTC traffic

intensity is considerably higher than at 0 UTC but the atmosphere is more convective and pollutant dispersion is enhanced.

In agreement with Duyzer et al. (2015) our modelling results show that Eixample and Gràcia-Sant Gervasi traffic stations do not represent the highest $NO_2$ concentrations in Barcelona. The highest levels are found in street canyons exposed to very high traffic intensity and not as well ventilated as the above-mentioned locations, and in open areas near highways surrounding the city. For example, measurements at 445 Valencia Street site show 20 % higher concentrations than in Eixample and Gràcia-Sant

Gervasi traffic sites on average during the morning peak on weekdays (see Fig. 8). Hence, additional monitoring sites located within highly trafficked streets are clearly needed to better represent highest $NO_2$ concentration levels in Barcelona.

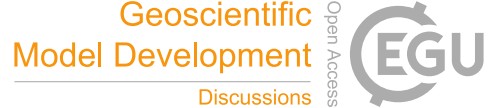



**Figure 11.** NO$_2$ high resolution (10 m × 10 m) concentration maps on 11th April 2013. a) represents concentrations at 0 UTC (2 am local time); b) is 7 UTC (9 am local time); c) is 12 UTC (2 pm local); and d) is 18 UTC (8 pm local).

## 5  Conclusions

This study describes the development of a coupled regional to street scale modelling system, CALIOPE-Urban, which provides high spatial and temporal resolution (up to 10 m × 10 m, hourly) NO$_2$ concentrations for Barcelona. It couples the mesoscale air quality forecasting system CALIOPE (WRF-HERMES-CMAQ-BSC-DREAM8b) with the urban roadway dispersion model,





R-LINE. For each regional 1 km × 1 km grid cell, meteorological data from WRF and background concentrations from CMAQ are used as input combined with traffic emissions from the HERMES emission model at road link level. R-LINE has been adapted to Barcelona's geometrical conditions by considering specific meteorology and background concentrations for each street. CALIOPE-Urban $NO_2$ simulations are compared with CALIOPE and with observations for temporal evaluation,

using data from five traffic sites and one urban background site during April and May 2013, and for spatial evaluation, with $NO_2$ concentrations measured by 182 passive dosimeters distributed across the entire city during two weeks in February-March 2017.

CALIOPE-Urban methodology adapts dynamically to street conditions by coupling the meteorology and background using street-specific surface roughness based on urban geometry. It adapts R-LINE dispersion model to compact cities using

channeled winds to drive dispersion and recalculated meteorological parameters for each street. Regarding background concentrations, it estimates over roof levels using an upwind background scheme and gives surface concentrations applying a vertical mixing parametrization based on urban geometry and atmospheric stability. The upwind background scheme avoids double counting traffic emissions in regional and dispersion models by using upwind grid cells concentrations to estimate over roof background concentrations. Doing so we omit the use of the grid cell over the estimated area, where traffic emissions are

considered in the dispersion model. To estimate background concentrations at surface level, the vertical mixing parametrization enhances background mixing from the overlying atmosphere under daytime convective atmospheric conditions and limits background air mixing during nighttime (stable) hours. For the transition from urban to suburban areas, CALIOPE-Urban implements a smooth variation for wind conditions, background and total concentrations.

Temporally, CALIOPE-Urban agrees better with observations than CALIOPE at the five traffic sites evaluated, where the

contribution of local emissions predominates. For the urban background site of Palau Reial, both systems give similar (good) results. For traffic sites, the coupled system shows better agreement in highly trafficked areas where local dispersion plays a crucial role. Regarding the diurnal average cycle in the observation sites, both systems follow the overall daily cycle in the observations but CALIOPE-Urban predicts better morning peaks, and corrects the afternoon levels at traffic sites as well as the systematic nighttime overestimation produced by the regional system. The vertical mixing of rooftop background concentra-

tions to surface levels based on atmospheric stability and urban geometry appears to be a good method to correct the strong positive bias of the mesoscale model under stable atmospheric conditions during the evening.

Spatially, CALIOPE-Urban performs better than CALIOPE at the dosimeters located close to traffic. This result is because R-LINE explicitly resolves road traffic emission dispersion simulating the high gradients of $NO_2$ observed levels that occur within a mesoscale system grid cell. CALIOPE-Urban gives more overestimation close to high trafficked areas. This behaviour

may be produced by an overestimation of traffic emissions in these roads or by underestimating dispersion. For dosimeters located more than 10 m away from traffic both systems perform reasonably well. The higher the traffic in the surrounding area, the better is CALIOPE-Urban performance compared to the regional system.

When exploring the main error sources, overall both systems produce results that are either accurate or inaccurate on the same days. This fact suggests that coupled system results are highly influenced by the regional system results. Furthermore, we

find that CALIOPE-Urban gives the higher errors (i.e. stronger underestimations) under stable conditions with light winds and




low PBL height than under more convective conditions, with stronger winds and higher PBL heights. Another potential source of uncertainty is the integration within HERMESv2.0 of COPERT IV instead of COPERT V, which considers diesel $NO_X$ exceedances derived from diesel-gate for EURO 5 and EURO 6 diesel cars (Brown et al., 2018). In a future work, we plan to update HERMESv2.0 with the new emissions factors released by COPERT V and examine the influence of traffic emissions in

CALIOPE-Urban results.

For high resolution air quality forecasts, we show that CALIOPE-Urban using either the numerical or the analytical dispersion algorithm gives good results. However, an entire city system execution using the analytical configuration takes approx. half the time compared to the numerical one. Hence, the analytical dispersion algorithm may be a suitable option for forecasting applications when sources, such as roadways, and receptors are located near the ground.

We show that traffic monitoring stations in Barcelona do not represent the highest $NO_2$ concentrations in the city. We find the highest levels in heavily trafficked street canyons that are not well ventilated and near highways in the city surroundings. As a consequence, we consider that additional monitoring sites located in these areas may better characterize the range of $NO_2$ concentration levels in Barcelona and give a better representation of human exposures.

This study has demonstrated that CALIOPE-Urban improves the accuracy of model outputs estimating $NO_2$ concentrations

in Barcelona compared to CALIOPE. The methodology is replicable in cities where a mesoscale chemistry transport model provides $NO_2$ simulations if urban geometrical data is available. The next step is to implement CALIOPE-Urban in the operational forecasting system for Barcelona to provide $NO_2$ concentrations at street level, and explore emissions impacts due to improved $NO_X$ emissions estimates.

*Code availability.* CALIOPE-Urban source code is available for non-commercial use. Contact Oriol Jorba (oriol.jorba@bsc.es) and Jaime

Benavides (jaime.benavides@bsc.es) for agreement details. Observational data in this work has been provided by co-authors from Institute of Environmental Assessment and Water Research, IDAEA-CSIC, Spain. Contact them if interested on these datasets.

**Appendix A:  Extended urban geometry characterization**





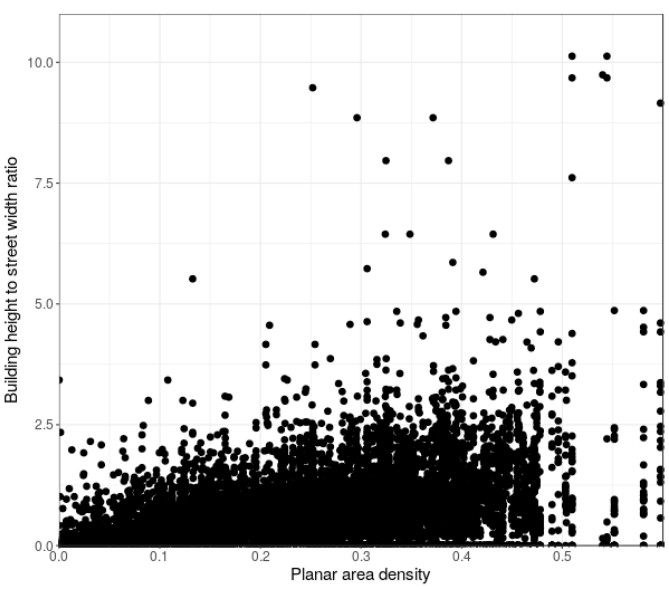

**Figure A1.** Scatter plot showing aspect ratio and building density relation in Barcelona city.

## Appendix B: Extended performance evaluation

**Table B1.** NO$_2$ model evaluation statistics calculated at six sites (described in Sect. 3) during April and May 2013 for CALIOPE and CALIOPE-Urban daily mean concentrations. Bold numbers represent model results with better performance for each statistic and site.

| Site | Method | FAC2 | MB | RMSE | GeoMean | GeoSD | r |
|---|---|---|---|---|---|---|---|
| 1. Palau Reial | CALIOPE | **1.00** | -10.97 | 15.44 | 1.30 | 1.08 | **0.84** |
| | CALIOPE-Urban | **0.95** | **-8.72** | **13.85** | 1.29 | **1.07** | 0.80 |
| 2. Eixample | CALIOPE | 0.91 | **-9.62** | 17.37 | 1.24 | 1.08 | 0.52 |
| | CALIOPE-Urban | **1.00** | 9.64 | **15.60** | 0.83 | **1.04** | **0.60** |
| 3. Gràcia-Sant Gervasi | CALIOPE | 0.91 | -10.97 | 15.44 | 1.30 | 1.08 | **0.82** |
| | CALIOPE-Urban | **1.00** | **-7.40** | **15.33** | 1.10 | **1.05** | 0.81 |
| 4. 213 Industria Street | CALIOPE | 0.75 | -19.15 | 23.51 | 1.62 | 1.20 | **0.68** |
| | CALIOPE-Urban | **0.95** | **-13.79** | **18.90** | 1.29 | **1.07** | **0.68** |
| 5. 455 Valencia Street | CALIOPE | 0.70 | -22.05 | 26.07 | 1.64 | 1.20 | **0.67** |
| | CALIOPE-Urban | **1.00** | **3.00** | **11.25** | 0.94 | **1.02** | 0.65 |
| 6. 309 Industria Street | CALIOPE | 0.95 | -7.13 | 13.33 | 1.22 | **1.07** | **0.78** |
| | CALIOPE-Urban | **0.95** | **-7.12** | **13.32** | 1.30 | **1.07** | 0.68 |





**Table B2.** NO$_2$ model evaluation statistics calculated at six sites (described in Sect. 3) during April and May 2013 for hourly concentrations of CALIOPE-Urban, CALIOPE-Urban Analytical or CALIOPE-Urban-nl (Non Local) configurations. Bold numbers represent model results with better performance for each statistic and site.

| Site | Method | FAC2 | MB | RMSE | GeoMean | GeoSD | r |
|---|---|---|---|---|---|---|---|
| 1. Palau Reial | CALIOPE-Urban | **0.72** | -8.70 | **21.57** | 1.28 | **1.22** | **0.57** |
| | CALIOPE-Urban Analytical | 0.69 | -10.46 | 22.44 | 1.38 | 1.24 | 0.56 |
| | CALIOPE-Urban-nl | 0.67 | **3.61** | 26.34 | **0.99** | 1.40 | 0.55 |
| 2. Eixample | CALIOPE-Urban | 0.86 | 9.38 | 26.70 | 0.83 | 1.12 | 0.55 |
| | CALIOPE-Urban Analytical | **0.87** | **7.99** | **26.25** | **0.85** | **1.11** | **0.56** |
| | CALIOPE-Urban-nl | 0.61 | 39.53 | 54.92 | 0.57 | 1.38 | 0.45 |
| 3. Gràcia-Sant Gervasi | CALIOPE-Urban | **0.79** | -7.39 | **25.11** | **1.07** | **1.19** | 0.52 |
| | CALIOPE-Urban Analytical | 0.78 | -9.29 | 25.54 | 1.13 | 1.20 | **0.53** |
| | CALIOPE-Urban-nl | 0.66 | **6.00** | 35.91 | 0.91 | 1.43 | 0.38 |
| 4. 213 Industria Street | CALIOPE-Urban | **0.78** | -13.62 | **26.65** | 1.30 | **1.17** | **0.57** |
| | CALIOPE-Urban Analytical | 0.76 | -14.85 | 27.22 | 1.35 | 1.18 | **0.57** |
| | CALIOPE-Urban-nl | 0.75 | **1.57** | 31.12 | **1.05** | 1.26 | 0.54 |
| 5. 455 Valencia Street | CALIOPE-Urban | 0.92 | 2.92 | 23.26 | 0.94 | **1.07** | 0.56 |
| | CALIOPE-Urban Analytical | **0.93** | **0.87** | **23.17** | **0.97** | **1.07** | **0.57** |
| | CALIOPE-Urban-nl | 0.79 | 23.72 | 42.29 | 0.75 | 1.19 | 0.47 |
| 6. 309 Industria Street | CALIOPE-Urban | **0.84** | **-4.60** | **22.72** | **1.05** | **1.13** | 0.53 |
| | CALIOPE-Urban Analytical | 0.83 | -6.64 | 22.94 | 1.12 | 1.14 | 0.54 |
| | CALIOPE-Urban-nl | 0.78 | 11.60 | 31.13 | 0.83 | 1.24 | **0.58** |

## Appendix C: Description of model evaluation statistics

Here we define the model evaluation statistics used to compare observed measurements (obs) with modelled concentrations (mod): the geometric mean bias (GeoMean), the fraction of model results within a factor of two of observations (FAC2), the geometric standard deviation (GeoSD), the correlation coefficient (R), the mean bias (MB) and the root mean square error (RMSE).

$$GeoMean = exp\left(\overline{\ln(obs)} - \overline{\ln(mod)}\right) \tag{C1}$$

$$FAC2 = 0.5 <= \frac{mod_i}{obs_i} <= 2.0 \tag{C2}$$

$$GeoSD = exp\left(\frac{\ln(F)}{\sqrt{2}\,eri^{-1}(A_F)}\right) \tag{C3}$$

$$R = \frac{\overline{(obs_i - \overline{obs})(mod_i - \overline{mod})}}{\sigma mod\,\sigma obs} \tag{C4}$$

$$MB = \frac{1}{n}\sum_{i=1}^{n} mod_i - obs_i \tag{C5}$$

$$RMSE = \sqrt{\frac{\sum_{i=1}^{n}(mod_i - obs_i)^2}{n}} \tag{C6}$$





where, $mod$ are modelled concentations; $obs$ are observed concentrations; overbar ($\bar{d}$) represents the average over a dataset $d$; $F$ is considered to be 2; $eri$ is the inverse of error function; $A_F$ is the proportion of the ratio; $\sigma d$ is the standard deviation of $d$; $n$ is the number of paired modelled and observed concentrations and subscripts represent a value between $one$ and $n$. For further details on the evaluation statistics we refer to Chang and Hanna (2004).

*Author contributions.*   Author contributions: JB developed the code. JB, MS, OJ designed the research. FA and XQ provided the observational data. CP, MG, FA, XQ and AS contributed to the discussion of the results. JB wrote the original paper, and all authors contributed to the review and editing of the paper.

*Competing interests.*   The authors declare that they have no conflict of interest.

*Disclaimer.*   This work was started while Dr. Michelle Snyder was a researcher at the University of North Carolina Institute for the Environ-
ment. Dr. Snyder's efforts throughout the project were completely voluntary.

*Acknowledgements.*   BSC researchers acknowledge the grants CGL2013-46736-R, CGL2016-75725-R and COMRDI15-1-0011-04 of the Spanish Government. J. Benavides PhD work is funded with the grant BES-2014-070637 from the FPI Programme by the Spanish Ministry of the Economy and Competitiveness. J. Benavides developed part of this work as research visitor at the Institute for the Environment at UNC funded with the mobility grant EEBB-I-17-12296 by the same Ministry. IDAEA-CSIC acknowledges the Barcelona City Council for
the support to the experimental campaign. Carlos Pérez García-Pando acknowledges long-term support from the AXA Research Fund, as well as the support received through the Ramón y Cajal programme (grant RYC-2015-18690) of the Spanish Ministry of Economy and Competitiveness.

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
