# Peer review of "CALIOPE-Urban v1.0: Coupling R-LINE with a mesoscale air quality modelling system for urban air quality forecasts over Barcelona city (Spain)"

_Geoscientific Model Development, 2019_

## Referee Comment (RC1) · Anonymous Referee #1 · 7 Mar 2019

**General comments**

The paper describes an interesting and innovative method for coupling the local air quality model R-LINE to a meso-scale air quality modelling system. It is generally clearly written and contains some useful analysis.

The analysis only considers NO2 concentrations. NO2 is a pollutant of active current interest for regulation and health effects, however it is challenging to model due to the interaction of dispersion and chemistry. Hence it is valuable to analyse modelled and

measured NOx concentrations before considering NO2, to assist with distinguishing between uncertainties in emissions, dispersion and chemistry. The consideration of chemistry effects should also be included in the associated discussions of NO2 results.

The method for taking into account the effects of a specific street canyon on dispersion described in Section 2.3.1 only considers flow channelling along the canyon. However, canyons are also known to cause recirculating flow across the canyon, which significantly alters the dispersion of road traffic emissions and hence the concentration variation with wind direction for receptors within the canyon. No analysis of the modelled or measured variation of concentrations with wind direction is presented, so it is difficult to assess the effectiveness of this formulation.

**Specific comments**

Section 2.1: Please state explicitly the depth of the lowest model layer in WRF and CMAQ, which is alluded to in Section 2.3.1.

Section 2.3.2 and Figure 3: Please comment on the negative value of intercept, which may indicate that the Ciutadella site does not fully represent an appropriate urban background concentration for the Eixample traffic site.

Section 2.3.3: Is the background mixing correction applied uniformly to all pollutants? In particular, O3 usually shows opposite behaviour to NOx and NO2, so this formulation may distort background concentrations used for chemistry calculations. Please also clarify how the background concentration is used within R-LINE, especially in regard to the chemistry calculations.

Section 2.4: Although the analytical model shows a significant reduction in execution time relative to the numerical local model, is 44 minutes execution time for 1 modelled hour realistic for use in an operational forecasting system?

Section 3.1: Please state the measurement height(s) for the official network sites.

Section 4: Please add an initial assessment of NOx modelled and measured concentrations.

Section 4.1: The discussion relating to Figure 8 does not mention the varying influence of chemistry processes through the day, which can lead to inaccuracies in diurnal profiles.

Section 4.3: It is common for Gaussian-type models such as R-LINE to perform poorly in low wind speed conditions due to uncertainties about associated wind directions. They also do not take into account possible accumulation of pollutants between hours in low wind speed conditions, which is in contrast to the assumption in the background adjustment in this work of reduced mixing causing reduced surface background concentrations. Figure 10f) suggests that the unadjusted regional background could be more appropriate than the adjusted in the early morning hours, though not in the evening. Are there other differences (eg. wind direction) between these two periods?

Figure 10: for panels e) and f) why is the model background concentration an average over six sites, not also taken from the Ciutadella site?

Section 5: Again, the uncertainty in NO2 resulting from chemistry processes should form part of the discussion.

**Technical corrections**

Abstract, p1 line 15: In this case, the coupled system **also** shows

Section 1, p3 line 8: subtract its result **from** the mesoscale model

Section 1, p3 line 30: please re-phrase 'over background roof-level concentrations' as the meaning is unclear

Section 1, p3 line 35 – p4 line 1: 5 traffic site**s** and 1 background

Section 1, p4 line 2: campaign that **deployed** 182 NO2 passive dosimeters **across Barcelona** for two weeks...

Section 2.3.3, p12 line 8: please re-phrase 'ends when the surface background gets over roof value for bd equals 0' as the meaning is unclear

Section 3.1, p15 line 8: centred **on** the measurement site

Section 4.1, p18 lines 9 and 11: unnecessary **the** before Appendix B

Figure 7: these plots look vertically distorted, as the target area is usually viewed as circular.

Figure 8: The vertical and horizontal axis scale labels are too small to read.

Section 4.1, p20 line 3: higher modelled traffic emissions, **resulting** in higher local pollutant concentrations...

Section 4.3, p23 lines 15-18: The first sentence says ten days of highest RMSE and ten days of lowest RMSE, whereas the following sentences suggest five days of high RMSE and five days of low RMSE. Please clarify how many days were selected and analysed.

Section 5, p27 line 11: ... gives surface concentrations **by** applying a vertical...

---

## Referee Comment (RC2) · Anonymous Referee #2 · 25 Mar 2019

This paper deals with the CALIOPE-Urban model system that couples regional scale atmospheric chemistry transport calculations with the urban roadway dispersion model R-LINE to represent the spatial distribution and temporal variability of NO2 concentrations in cities. A new concept to adapt R-LINE to dense urban areas is introduced, consisting of an urban background scheme, a vertical mixing parametrization considering atmospheric stability and building geometries and a surface roughness parametrization to estimate turbulent dispersion in street canyons. The paper is clearly structured and the new concept is well documented for the most part. Results from the model eval-

uation with different monitoring datasets obtained in Barcelona are convincingly presented. Specifically, the coupled system overcomes two major challenges of the downscaling of regional model results to local scales, namely the adjustment of mesoscale meteorological data to street level and the retrieval of background concentrations from the regional model avoiding double counting of traffic emissions. My main concerns are with regard to two aspects of the new concept. First, it is claimed that atmospheric stability influences the relation between rooftop concentrations and the concentrations at street level (e.g. p. 11, lines 8-11). While this well may be the case, no independent evaluation of the influence of atmospheric stability on the vertical concentration profile in street canyons is given in the paper. The authors should either refer to previous studies in Barcelona or show an evaluation based on own measurement data. Second, the model system shows poor skill when the observed wind speed is low. I would expect that the traffic-induced turbulence dominates the turbulence in street canyons at low wind speeds. However, it seems that turbulence generated by the moving traffic is not included in the parametrization. Calm winds potentially lead to highest concentrations and can cause severe pollution episodes. Hence, it would be crucial for a street canyon model to cope with low wind situations.

Specific Comments

1.) P. 2 line 9-21: In this part of the Introduction, several systems coupling regional and urban scale models are described. It would be better to divide this presentation into (1) systems that apply nesting of an urban scale model within a regional scale model and (2) regional scale models that apply downscaling (using a dispersion kernel). The given examples from literature are not exhaustive. Also mention a second method for downscaling, by embedding Gaussian dispersion models within the grid.

2.) P.4 line 1: How representative is this period (April and May 2013) for the season? Why was such a short period chosen?

3.) P.5 line 26-27: Why were the 38 vertical layers from WRF collapsed to 15 layers for

the CMAQ computation? With only six layers in the PBL, this leads to a rather crude treatment of the near-ground chemistry and boundary layer mixing processes.

4.) P.4 line 1: Please provide a list of the chemical reactions in the GRS here.

5.) P. 7 line 3 and P.12 line 4: Several of the empirical parametrizations in this paper have been calibrated with NO2 measurements (parameters C and m). This raises the question about independence of the calibration data. Was the calibration done with an independent NO2 dataset, not used in the presented model evaluation?

6.) P.7 line 25: Wind channelling may not occur in streets that are relatively short. The validity of the channelling effect should be analysed for street network of Barcelona.

7.) Section 2.3.3: The large scale model grids are step wise in nature. This could lead to significant edge effects caused by the concentration steps between the CMAQ grid cells. How is this considered in the UBS when applying bilinear interpolation to provide background concentrations at the receptors? The error of the background concentrations at low wind speeds should be estimated.

8.) P. 11 line 10: Does wind channelling affect the ratio ws_sfc/ws_bh?

9.) P.12 line 5 - 10: Determination of the surface background concentrations needs more explanation. An illustration of the surface background concentration as function of building density would be helpful for understanding how it is derived from the rooftop background under different stability conditions.

10.) P.15 line 11 - 12: Which QA/QC procedure was in place for the monitoring with passive dosimeters?

11.) Table 3 and Figure 8: Measurements at station Gracia-Sant Gervasi are underestimated by all three model configurations in the daytime between morning and afternoon rush hours. Table 3 shows a positive bias for CALIOPE-urban-nl (marked as best performance for MB at this site), but this is deceiving since Figure 8 shows that overestimation at rush hours increased the bias. Obviously, the traffic increment is not

correctly represented at this site. Could this be caused by the missing contribution from recirculation of traffic exhaust?

12.) P.19 line 9 and Figure 8: Give the possible reason for the afternoon underestimation of NO2 concentrations at sites with low traffic. The underestimation of NO2 in the afternoons could also be linked to photochemical conversion. Therefore, I recommend to repeat the plots of Figure 8 for NOx concentrations.

Technical Corrections

P. 6 line 8: "This approach addresses" fits better.

P.15 line 12: In every km2 grid cell?

Figure 11: It should be mentioned in the figure caption whether the resolution is 10m x 10m for the entire concentration map or only in the 250m buffers along streets.

Figure A1: What explains the zero values for the aspect ratio values in the scatterplot?

---

## Author Comment (AC1) · 28 May 2019

**Point by point responses to the Anonymous Referee #1**

We are very thankful to the reviewer for providing a detailed revision of our manuscript. The comments of the reviewer are indicated point-by-point in the following text. We explain how we have carefully addressed each of them (our answers in blue text). Modifications and new sections are highlighted with track changes in the manuscript and the Supporting Information.

**General comments**

**1. Reviewer #1.** The analysis only considers $NO_2$ concentrations. $NO_2$ is a pollutant of active current interest for regulation and health effects, however it is challenging to model due to the interaction of dispersion and chemistry. Hence it is valuable to analyse modelled and measured $NO_X$ concentrations before considering $NO_2$, to assist with distinguishing between uncertainties in emissions, dispersion and chemistry. The consideration of chemistry effects should also be included in the associated discussions of $NO_2$ results.

**Authors:**

With regard to the chemistry used in the street-scale model, Valencia et al. (2018) evaluated multiple $NO$-$NO_2$-$O_3$ chemical mechanisms in R-LINE. They compared the GRS mechanism, used by ADMS-Urban (Malkin et al., 2016), and two other algorithms for $NO_X$ chemistry with near-road data in Michigan. Their results indicate that the GRS mechanism was the most consistent in predicting $NO_2$ for near-roadway environments. We believe, based on the Valencia et. al. (2018) evaluation of GRS in R-LINE and the use of GRS in the ADMS-Urban model, that this is an appropriate way to model $NO_2$ chemistry without the need to also evaluate $NO_X$.

However, following the advice of the reviewer we show below the comparison of $NO_X$ average daily cycle concentrations for each model (equivalent figure as Figure 8 of the initial submitted manuscript). We can see that the errors for $NO_2$ and $NO_X$ are very similar for all the stations, indicating that the chemistry of GRS is performing well in our simulation experiments. Thus, we believe that it is reasonable to evaluate modelled $NO_2$ directly as suggested by Valencia et. al. (2018) and Malkin et al. (2016). Given that the paper is focused on $NO_2$ due to its interest for regulation and health effects we decided not to include the $NO_X$ discussion in the manuscript and keep the focus on the $NO_2$ exclusively.

[Figure]

**Figure 1. NO$_X$ average daily cycle at all sites described in Sect. 3.1 during April and May 2013 for weekday and weekend. Observations are represented in black coloured lines, red lines are CALIOPE, blue lines are CALIOPE-Urban and green lines represent CALIOPE-Urban without local developments (CALIOPE-Urban-nl).**

**2. Reviewer #1.** The method for taking into account the effects of a specific street canyon on dispersion described in Section 2.3.1 only considers flow channelling along the canyon. However, canyons are also known to cause recirculating flow across the canyon, which significantly alters the dispersion of road traffic emissions and hence the concentration variation with wind direction for receptors within the canyon. No analysis of the modelled or measured variation of concentrations with wind direction is presented, so it is difficult to assess the effectiveness of this formulation.

**Authors:**

We agree with the reviewer on the importance of modelling the recirculating flow across the canyon and its impact on pollutant dispersion. It is widely known that canyons with sufficient aspect ratio (i.e. more than 0.15) may cause recirculating flow across the canyon (Oke, 1988; Yamartino and Wiegand., 1986; Dobre et al., 2005) when over roof winds are close to perpendicular to street direction. This recirculation may affect significantly the dispersion of road traffic emissions. However, in this work we assume that recirculation process is negligible for multiple reasons: first, dispersion models, such as R-LINE, are not designed to model extremely detailed urban flows (e.g. CFD models), but rather are based on representative flows that are influenced by average urban attributes near the source location; second, dispersion models are designed to give accurate concentrations averaged over a long a time period (usually one hour) where variability in wind speeds occur and thus recirculation may not be longer consistent; third, there is a recirculation and a vehicle induced turbulence occurring within the street canyon, both are contributing to a well mixed more homogeneous air mass within the street canyon, especially over the long averaging time and variable wind conditions; and lastly evaluation of the potential impact (positive or negative) of including recirculating flows across the canyon is not possible without multiple simultaneous meteorological and pollutant measurements within a street canyon at a fine temporal scale, which was not part of the experimental design and thus are not available.

We have added an explicit note about this limitation in the revised manuscript in in Section 2.3.1, page 8 lines 18-22 as "In this work, we assume that recirculation flows within street canyons are negligible because R-LINE computes concentrations averaged over an hour, when recirculation and vehicle induced turbulence are assumed to contribute to a well mixed more homogeneous air mass driven by variable wind conditions. Additionally, evaluation of the potential impact of including recirculating flows across the canyon is not possible without multiple simultaneous meteorological and pollutant measurements at a fine temporal scale, which are not available.".

Regarding flow channelling along canyon, we didn't assess the effectiveness of this formulation because we don't have access to a complete dataset of measured wind conditions within a diverse range of streets in the city. However, the positive results of the work indicate that our formulation is appropriate for the objective of the study.

**Specific comments**

**3. Reviewer #1.** Section 2.1: Please state explicitly the depth of the lowest model layer in WRF and CMAQ, which is alluded to in Section 2.3.1.

**Authors:**

We have added a comment in Section 2.3.1, page 6 lines 19-21 as "Most buildings in Barcelona have lower heights than the WRF bottom layer (40.6 m depth). WRF results are assumed to represent over roof wind and stability conditions because its mid-point height (20.3 m) is similar to average building height (bh) in a typical neighbourhood of Barcelona (e.g. Eixample district; 20.7 m)."

**4. Reviewer #1.** Section 2.3.2 and Figure 3: Please comment on the negative value of intercept, which may indicate that the Ciutadella site does not fully represent an appropriate urban background concentration for the Eixample traffic site.

**Authors:**

We believe that Ciutadella is a reasonable background site due to its upwind location in the predominant wind direction during the day and its location within the main park of the city (see Figure 2 in the manuscript). In addition, during the two-week period of the passive dosimeters campaign its mean $NO_2$ concentrations were 40.2 $\mu g/m^3$, which is very close to the observed mean concentrations of 42.1 $\mu g/m^3$ using the passive dosimeters, suggesting that it is a reasonable background site (Amato, personal communication, April 24, 2019). The value of the intercept is very close to zero (i.e. the remaining background influence), which means that the regional and urban background contribution have been taken out reasonably well.

**5. Reviewer #1.** Section 2.3.3: Is the background mixing correction applied uniformly to all pollutants? In particular, $O_3$ usually shows opposite behaviour to NOx and $NO_2$, so this formulation may distort background concentrations used for chemistry calculations. Please also clarify how the background concentration is used within R-LINE, especially in regard to the chemistry calculations.

**Authors:**

Yes, the background vertical mixing is applied uniformly as done in all split-operator models. The vertical distribution of pollutants are solved first with a vertical diffusion following similarity theory, applied uniformly to all pollutants, and then we solve the chemistry. With this approach, $O_3$ shows opposite profile to $NO_X$ and $NO_2$ as noted by the reviewer. This is consistent with the chemical reaction of emitted NO with ambient $O_3$ to form $NO_2$. We have included a clarifying note in the revised manuscript in Section 2.3.3, page 11 lines 10-12 as "To calculate street-level $NO_2$ concentrations, the vertical distribution of pollutants are solved first using the background decay method, applied uniformly to all pollutants, and then the GRS chemical mechanism is solved."

**6. Reviewer #1.** Section 2.4: Although the analytical model shows a significant reduction in execution time relative to the numerical local model, is 44 minutes execution time for 1 modelled hour realistic for use in an operational forecasting system?

**Authors:**

The current model design and methodology was explored as a potential way to forecast pollutant levels in the future, therefore we are using this initial evaluation to determine if this is possible and with what level of accuracy. It is important to explore both the analytical and numerical calculations in R-LINE to determine strengths and weaknesses of both. Here we present that the analytical solution is much faster, however the numerical solution is more accurate in some instances, so a final forecasting model would need to balance speed and accuracy. Once, we determine the validity and accuracy of our method we will begin the process of code optimization. For instance, the R-LINE code is not currently parallelized. Parallelization could be done at the road-segment level, which will speed-up the code by several orders of magnitude, making this an extremely cheap and valuable tool for a forecasting system.

**7. Reviewer #1.** Section 3.1: Please state the measurement height(s) for the official network sites.

**Authors:**

We added a note in Section 3.1 page 15 Table 3 to clarify this: "The measurement height of the official network sites and the mobile sites is 3 meters".

**8. Reviewer #1.** Section 4: Please add an initial assessment of $NO_X$ modelled and measured concentrations.

**Authors:**

As explained in response number 1, R-LINE has been evaluated for roadways in urban areas with inert pollutants (such as $NO_X$) and reactive pollutants (such as $NO_2$), therefore there is no value added to present a full evaluation of $NO_X$ in this instance. In addition, chemical transport models such as CMAQ have been evaluated for use in urban areas and have been previously used to provide background concentrations (Beevers et al., 2012; Isakov et al., 2014). We are using these previous models as is, by coupling R-LINE and CMAQ, and making adjustments based on the data to evaluate the additions of the street canyon and background adjustments. The street canyon adjustments are evaluated using a variety of street canyons throughout Barcelona. The background adjustments are evaluated at background sites throughout the city.

Considering that the scope of the present manuscript is the modelling of $NO_2$ concentrations at street level, we prefer not to add the discussion of $NO_X$.

**9. Reviewer #1.** Section 4.1: The discussion relating to Figure 8 does not mention the varying influence of chemistry processes through the day, which can lead to inaccuracies in diurnal profiles.

**Authors:**

R-LINE first calculates the dispersion of pollutants from the road source, and then it resolves the parameterised equations of the chemical reactions for the pollutant transportation time interval. The GRS chemistry mechanism solves the photochemistry of $NO_2$ assuming clear-sky conditions. Hence, it does not consider cloud effects on photolysis, representing one of its major limitations.

Valencia et. al. (2018) also show that the GRS method in R-LINE has less than a 15% bias of the results even though they do not account for cloud cover. From our experience, the processes that may have a greater influence on the results in this modelling system are the correct wind and stability conditions and the accuracy of emission sources within the street canyons. These may have more influence on the concentrations than the photolysis in the chemistry scheme. For example, in Figure 2 we show the weekday average daily cycle for Valencia Street 445 to compare the effect of setting the GRS mechanism photolysis rates to zero (caliope_urban_no_photo) with the effect of setting atmospheric conditions to stable (caliope_urban_stable). The stable conditions are set using the following parameter values from Snyder et al. (2013): Lmon (Monin-Obukhov length) equals 11.1; ustar (friction velocity) equals 0.12; Wsrefh (wind speed at roof-top level) equals 2.0. The impact of neglecting completely the photochemical reaction of the GRS chemical mechanism results in an overestimation of NO2 concentrations during daytime. Although we see a negative impact of not using the photochemical reaction in the GRS chemical mechanism (purple line), it is clear that setting stable atmospheric conditions dramatically changes the modelled concentration levels, producing a greater overestimation (green line). These results confirm our initial hypothesis that atmospheric stability has a much greater influence on the $NO_2$ concentration than neglecting clouds in the calculation of the $NO_2$ photolysis rate applied in the GRS chemical mechanism.

[Figure]

**Figure 2. NO₂ average daily cycle at Valencia Street 445 site described in Sect. 3.1 during April and May 2013 for weekday. Observations are represented in black coloured lines, red lines are CALIOPE, blue lines are CALIOPE-Urban, green lines represent CALIOPE-Urban with stable atmospheric conditions (caliope_urban_stable) and purple lines depict CALIOPE-Urban with photolysis rates set to zero (caliope_urban_no_photo).**

In the revised manuscript, we have added a note about neglecting the effect of clouds in the R-LINE photolysis rate (Section 2.2, page 6 lines 3-4): "GRS chemistry mechanism solves the photochemistry of NO₂ assuming clear-sky conditions. Thus, it does not consider cloud effects on the NO₂ photolysis rate, representing this one of its major limitations."

**10. Reviewer #1.** Section 4.3: It is common for Gaussian-type models such as R-LINE to perform poorly in low wind speed conditions due to uncertainties about associated wind directions. They also do not take into account possible accumulation of pollutants between hours in low wind speed conditions, which is in contrast to the assumption in the background adjustment in this work of reduced mixing causing reduced surface background concentrations. Figure 10f) suggests that the unadjusted regional background could be more appropriate than the adjusted in the early morning hours, though not in the evening. Are there other differences (eg. wind direction) between these two periods?

**Authors:**

Regarding the accumulation of pollutants, it's true that we can't consider it from one hour to the next within the street canyon. As the reviewer correctly identifies, this is a limitation in dispersion models and in our implementation.

Regarding the reviewer comment on Figure 10 panel f) that the unadjusted regional background could be more appropriate than the adjusted in the early morning hours, we are aware that the result from the upwind background scheme (assumed as roof-level background concentration provider) gives a more precise result from 0 to 7 UTC under calm winds in Figure 10 panel f). In contrast, at night from 18 to 23 UTC the opposite happens, background concentrations at surface level are more accurate than at roof level in comparison with observations. We see this result as positive because it suggests that our method can reduce the overestimation of night surface level concentrations. The contradictory result at surface level between morning and night hours for stable hours with calm wind conditions is dependent on the results of the mesoscale system and we would need more observational data to further investigate this issue.

Concerning the uncertainties associated to wind direction under low wind conditions, we show in the following figure the difference in wind direction at the Barcelona airport for the two periods discussed in Section 4.3:

[Figure]

**Figure 3. Boxplots by time of the day of good performance days for WRF (blue) and observed (orange) wind directions at Barcelona airport (10m height) with dots representing outliers.**

[Figure]

**Figure 4. Boxplots by time of the day of bad performance days for WRF (blue) and observed (orange) wind directions at Barcelona airport (10m height) with dots representing outliers.**

From the two figures, we see a similar pattern in both periods in general, being the exception the morning transition from stable to unstable atmospheric conditions (5-8 UTC) in bad performance days. The wind direction difference between WRF and observations is approximately 90 degrees on average. In contrast, looking at Figure 10 panel d) (surface $NO_2$ concentrations) and f) (background $NO_2$ concentrations), we do not see a clear impact of the mentioned error of wind direction from 5 to 8 UTC. In our system, wind direction may not play a crucial role when wind speed is very low (i.e. for bad performance days approx. 1.5 m/s) for two reasons: (1) the upwind background scheme will take CMAQ concentrations from nearby grid cells (i.e. tending to give similar concentration levels in different wind directions), typically within city boundaries, and (2) R-LINE meandering partial contribution, which disperses radially in all directions, is greater under low wind speeds reducing the potential impact of the wind direction error.

**11. Reviewer #1.** Figure 10: for panels e) and f) why is the model background concentration an average over six sites, not also taken from the Ciutadella site?

**Authors:**

As stated in response to reviewer comment 4, the Ciutadella Park monitoring station is the only reference site available for urban background in Barcelona. Our aim in Figure 10 for panels e) and f) is to compare the modelled background concentrations (i.e.

excluding local vehicular traffic contribution) with the most suitable observed urban background, which in our case is given by Ciutadella site. The model background concentration is taken to be an average over six sites because the interest is to represent a summary of the variability of the modelled background at the six sites in comparison with the observed background. We added a clarification in the revised manuscript in Section 4.3 page 25 lines 3-4: "We aimed to compare the modelled background concentrations (i.e. excluding local vehicular traffic contribution) with the most representative urban background observation, which in our case is the Ciutadella site."

**12. Reviewer #1.** Section 5: Again, the uncertainty in $NO_2$ resulting from chemistry processes should form part of the discussion.

**Authors:**

We added a comment about the uncertainty of chemistry processes used in our solution in Section 5, page 28 lines 6-7: "Finally, we consider an additional source of uncertainty the assumption of clear-sky conditions in the photolysis rate calculation of the GRS chemistry mechanism."

**Technical corrections**

**13. Reviewer #1.** Abstract, p1 line 15: In this case, the coupled system **also** shows

**Authors:** Amended

Section 1, p3 line 8: subtract its result **from** the mesoscale model

**Authors:** Amended

Section 1, p3 line 30: please re-phrase 'over background roof-level concentrations' as the meaning is unclear

**Authors:** Amended

Section 1, p3 line 35 – p4 line 1: 5 traffic site**s** and 1 background

**Authors:** Amended

Section 1, p4 line 2: campaign that **deployed** 182 NO2 passive dosimeters **across Barcelona** for two weeks..

**Authors:** Amended

Section 2.3.3, p12 line 8: please re-phrase 'ends when the surface background gets over roof value for bd equals 0' as the meaning is unclear

**Authors:** Amended

Section 3.1, p15 line 8: centred **on** the measurement site

**Authors:** Amended

Section 4.1, p18 lines 9 and 11: unnecessary **the** before Appendix B

**Authors:** Amended

Figure 7: these plots look vertically distorted, as the target area is usually viewed as circular.

We agree with the reviewer that the image looks vertically distorted but the current version of the Delta Tool for Windows is producing this kind of plot and as far as we know we can't do anything to change it. We downloaded the most updated version and it produced the same kind of plot. In the informational website of Delta Tools it is shown as vertically distorted, too: https://ec.europa.eu/jrc/en/scientific-tool/fairmode-delta-tool

Figure 8: The vertical and horizontal axis scale labels are too small to read.

**Authors:** Amended

Section 4.1, p20 line 3: higher modelled traffic emissions, **resulting** in higher local pollutant concentrations...

**Authors:** Amended

Section 4.3, p23 lines 15-18: The first sentence says ten days of highest RMSE and ten days of lowest RMSE, whereas the following sentences suggest five days of high RMSE and five days of low RMSE. Please clarify how many days were selected and analysed.

**Authors:** Amended

Section 5, p27 line 11: ... gives surface concentrations **by** applying a vertical…

**Authors:** Amended

**References**

Beevers, S., Kitwiroon, N., Williams, M.L., and Carslaw, D.C. "One way coupling of CMAQ and a road source dispersion model for fine scale air pollution predictions". Atmospheric Environment 59, pp. 47–58. DOI: https://doi.org/10.1016/j.atmosenv.2012.05.034. 2012.

Dobre, A., Arnold, S. J.Smalley, R. J., Boddy, J. W D, Barlow, J. F. "Flow field measurements in the proximity of an urban intersection in London, UK". Atmospheric Environment 39, pp. 4647-4657. DOI: https://doi.org/10.1016/j.atmosenv.2005.04.015. 2005.

Isakov, V. et al. "Air Quality Modeling in Support of the Near-Road Exposures and Effects of Urban Air Pollutants Study (NEXUS)". International Journal of Environmental Research and Public Health 11.9, pp.

8777–8793. DOI: https://doi.org/10.3390/ijerph110908777. 2014.

Malkin, T.L., Heard, D.E., Hood, C., Stocker, J., Carruthers, D. "Assessing chemistry schemes and constraints in air quality models used to predict ozone in London against the detailed Master Chemical Mechanism". Faraday Discussions 189, pp. 589-616. DOI: https://doi.org/10.1039/C5FD00218D. 2016.

Oke, T.R. "Street design and urban canopy layer climate". Energy and Buildings 11.1-3, pp. 103–113. DOI: https://10.1016/0378-7788(88)90026-6. 1988.

Valencia, A., Venkatram, A., Heist, D., Carruthers, D., and Arunachalam, S. "Development and evaluation of the R-LINE model algorithms to account for chemical transformation in the near-road environment". Transportation Research Part D: Transport and Environment 59.2, pp. 464–477. DOI: https://doi.org/10.1016/j.trd.2018.01.028. 2018.

Yamartino, R.J., Wiegand, G. "Development and evaluation of simple models for flow, turbulence and pollutant concentration fields within an urban street canyon". Atmospheric Environment 20, pp. 2137-2156. 1986.

---

## Author Comment (AC2) · 28 May 2019

**Point by point responses to the Anonymous Referee #2**

We are very thankful to the reviewer for providing a detailed revision of our manuscript. The comments of the reviewer are indicated point-by-point in the following text. We explain how we have carefully addressed each of them (our answers in blue text). Modifications and new sections are highlighted with track changes in the manuscript and the Supporting Information.

**General comments**

**1. Reviewer #2.** First, it is claimed that atmospheric stability influences the relation between rooftop concentrations and the concentrations at street level (e.g. p. 11, lines 8-11). While this well may be the case, no independent evaluation of the influence of atmospheric stability on the vertical concentration profile in street canyons is given in the paper. The authors should either refer to previous studies in Barcelona or show an evaluation based on own measurement data.

**Authors:**

Several dispersion models integrate in their formulation the concept of street and over-roof concentrations exchange dependent on atmospheric stability (Hotchkiss and Harlow., 1973; Berkowicz et al., 2000; Soulhac et al., 2011; Kim et al., 2018). In addition, this influence has been demonstrated using wind tunnel experiments (Salizzoni et al., 2009). The influence of atmospheric stability on street wind speed and over roof winds has been shown using experimental measurements, too (Rotach,. 1995). For these reasons, we consider scientifically robust to assume that atmospheric stability influences the relation between rooftop concentrations and the concentrations at street level.

The methodology in this manuscript is a first attempt at coupling the urban-scale model, R-LINE, with a mesoscale model, CMAQ, to obtain concentrations throughout the city. We do present multiple street level measurements of varying degrees of urban structural influence. The results indicate that using one consistent background methodology in all these instances provides validity in our approach. Our methodology is still under refinement and will need further evaluation based on additional datasets. We are currently working on the analysis of measured vertical profiles of Black Carbon (BC) within a few street canyons of the Barcelona city. Unfortunately, we don't have access to high frequency vertical profiles of $NO_2$ concentrations and wind conditions within street canyons in Barcelona. From BC vertical profiles results, it seems that our hypothesis is well-oriented as the reviewer can see in the figure below, where we show the modelled contributions compared with the hourly averaged observed BC at different heights in a very narrow Street in Barcelona. We show 12 UTC (13 hour local time), an

hour that it is expected to have a low contribution from local traffic (i.e. more signal from background). In addition, we expect a well mixed vertical BC column due to the convective conditions typically occurring at this period of the day. We see in the figure that the overall dynamic of the modelled vertical profile is in agreement with the observed profile.

[Figure]

**Figure 1. Local traffic and background contributions to BC vertical profile at Torrent de l'Olla Street at 12 UTC on 20th November 2015. Observations are depicted as dots and coloured levels represent each local traffic contribution: nearest roads is light blue, roads within 191 m (excluding the nearest roads) is dark blue, roads in 392 m (excl. roads within 191 m) is light green, dark green is for roads in 1 km$^2$ (excl. roads within 392 m) and pink is for background.**

Lastly, it is also important to note that there may not be datasets available that specifically address all possible aspects of our approach, and thus we must rely on the datasets available and on the results of our methodology within the modeling system.

We added a note in Section 2.3.3 page 11 lines 14-16 to support the incorporation of atmospheric stability influence on vertical mixing: "In the research literature, the influence of atmospheric stability on vertical mixing within a street canyon has been demonstrated using experimental measurements (Rotach, 1995), wind tunnel experiments (Salizzoni et al., 2009), and it has been implemented in some dispersion models (e.g. Soulhac et al., 2011; Kim et al., 2018)".

**2. Reviewer #2.** Second, the model system shows poor skill when the observed wind speed is low. I would expect that the traffic-induced turbulence dominates the turbulence in street canyons at low wind speeds. However, it seems that turbulence generated by the moving traffic is not included in the parametrization. Calm winds potentially lead to highest concentrations and can cause severe pollution episodes. Hence, it would be crucial for a street canyon model to cope with low wind situations.

**Authors:**

In our approach, we have not considered the traffic-induced turbulence directly, however we do have an initial vertical dispersion of roadway emissions which somewhat models traffic-induced turbulence. In previous roadway studies (Snyder et al., 2013; Heist et al., 2013) this same approach has been used in the median of the roadway for Caltrans Highway 99 (Benson, 1989) and results are accurate when compared to near-road measurements, therefore not explicitly modeling vehicle-induced turbulence is not believed to have a large impact on the results.

**Specific Comments**

**3. Reviewer #2.** 1.) P. 2 line 9-21: In this part of the Introduction, several systems coupling regional and urban scale models are described. It would be better to divide this presentation into (1) systems that apply nesting of an urban scale model within a regional scale model and (2) regional scale models that apply downscaling (using a dispersion kernel). The given examples from literature are not exhaustive. Also mention a second method for downscaling, by embedding Gaussian dispersion models within the grid.

**Authors:**

In the revised manuscript we present a more complete list of systems that couple off-line regional and urban scale models by downscaling the regional model using a dispersion kernel. We consider adequate to uniquely present downscaling methods because our system belongs to this category, which as far as we know is the most extended methodology to couple regional and urban scales. The revised paragraph in the manuscript is as follows (Section 1 Page 2 Line 9 to 27),

"In order to overcome these limitations, coupling off-line the regional and urban scales by downscaling the regional model using a dispersion kernel has been successfully applied in some cities (Beevers et al., 2012; Moussafir et al., 2014; Isakov et al., 2014; Jensen et al., 2017; Maiheu et al., 2017; Kim et al., 2018; Hood et al., 2018, Fagerli et al., 2019). For instance, Hood et al. (2018) coupled a regional climate-chemistry model with 5 km horizontal resolution (EMEP4UK) with the fine-scale model ADMS-URBAN to simulate air quality over London in 2012. They compared the coupled system results with the regional and the fine-scale models run separately. Authors found that both the fine-scale model and the coupled system performed better than the regional for $NO_2$ at both annual mean and hourly concentration levels due to the explicit treatment of traffic emissions within the city. In addition, Jensen et al. (2017) estimated annual $NO_2$ concentrations at 2.4 million addresses in Denmark using the street canyon model OSPM coupled with DEHM for regional background concentrations and UBM for urban background obtaining a good correlation in Copenhaguen ($r^2$ = 0.70) against 98 measurement sites for $NO_2$ in the year 2012. Maiheu et al. (2017) covered a broader

spatial context, estimating EU-wide $NO_2$ annual average levels at 100 meter resolution with a regional model coupled with a dispersion kernel-based method. The approach does not produce hourly concentration levels and approximates road-link level traffic emissions by distributing the regional grid cell traffic emissions to each road-link based on road capacity. Hence, it provides more spatial detail than previous EU scale $NO_2$ assessment studies, but more specific methods are required to resolve air quality in cities. In this sense, there is a lack of air quality urban forecasting methodologies that can be applied to a diverse range of cities and that consistently resolve at least some of the major challenges already identified by the community, i.e., 1) downscaling regional meteorology to street level as required to drive pollutant dispersion; 2) obtaining background concentrations from the mesoscale system avoiding the double counting of traffic emissions. Additionally, we consider vertical mixing with background air a key process to be resolved when coupling the regional and urban scales."

**4. Reviewer #2.** 2.) P.4 line 1: How representative is this period (April and May 2013) for the season? Why was such a short period chosen?

**Authors:**

$NO_2$ exceedances in BCN are chronic along the year. April and May are months with a reduced amount of holidays and vehicular traffic behaviour is representative of the pulse of the city. We are aware that a longer period would be of interest to evaluate the skills of the model across different seasons. This will be presented in a future work. In this work, we focus on the experimental campaign with multiple simultaneous measurements along trafficked street canyons. We find this dataset relevant because we can evaluate CALIOPE-Urban close to road sources.

**5. Reviewer #2.** 3.) P.5 line 26-27: Why were the 38 vertical layers from WRF collapsed to 15 layers for the CMAQ computation? With only six layers in the PBL, this leads to a rather crude treatment of the near-ground chemistry and boundary layer mixing processes.

**Authors:**

In Europe, there are a wide range of mesoscale air quality models that work with low vertical resolution for computational reasons (e.g., LOTOS-EUROS, CHIMERE), and the skills of those models have been shown to be in the same order as other systems with higher vertical resolution. We use the default CALIOPE forecast configuration, which aims to reduce computational time to allow for rapid forecasting. CALIOPE skills are within the state-of-the-art forecasting systems (e.g. Pay et al., 2014). Since we are

most concerned with $NO_2$ here, which has a rapid chemical transition from emitted NO to ambient $NO_2$, the most important chemistry is the near-road chemistry that is simulated in the fine-scale dispersion model.

**6. Reviewer #2.**4.) P.4 line 1: Please provide a list of the chemical reactions in the GRS here.

**Authors:**

We have included a list of the chemical reactions in the GRS in Section 2.2 page 6 Table 1 and a note referencing the table in page 6 lines 1-3: "In order to estimate $NO_2$ concentrations, R-LINE incorporates a chemistry module to resolve simple NO to $NO_2$ chemistry with the Generic Reaction Set (GRS; Valencia et al., 2018) considering the chemical reactions in Table 1."

**7. Reviewer #2.** 5.) P. 7 line 3 and P.12 line 4: Several of the empirical parametrizations in this paper have been calibrated with $NO_2$ measurements (parameters C and m). This raises the question about independence of the calibration data. Was the calibration done with an independent $NO_2$ dataset, not used in the presented model evaluation?

**Authors:**

The scarcity of measurements didn't allow us to separate the observations for an independent calibration and validation process. We have used the whole set of observations to calibrate and evaluate the model. As responded in reviewer comment 4, we are aware that a longer period with an independent $NO_2$ dataset would be of interest to evaluate the skills of the model across different seasons. This will be presented in a future work.

**8. Reviewer #2.** 6.) P.7 line 25: Wind channelling may not occur in streets that are relatively short. The validity of the channelling effect should be analysed for street network of Barcelona.

**Authors:**

We tried to apply a common simple approach for the entire city. We agree with the reviewer that a more refined implementation of the channelling will be needed in the future but it is out of the scope of the present paper. We didn't assess the effectiveness of the channelling effect formulation because we don't have access to a complete dataset of measured wind conditions within a diverse range of streets in the city.

**9. Reviewer #2.** 7.) Section 2.3.3: The large scale model grids are step wise in nature. This could lead to significant edge effects caused by the concentration steps between the CMAQ grid cells. How is this considered in the UBS when applying bilinear interpolation to provide background concentrations at the receptors? The error of the background concentrations at low wind speeds should be estimated.

**Authors:**

We estimate background concentrations at roof level using the urban background scheme in two steps. First, our method selects CMAQ cells as background concentration providers depending on the wind speed and direction. Second, for each receptor we apply a bilinear interpolation method to provide a background at very high resolution calculating weights at each receptor and computing weighted data.

With regard to the comment "The Error of background concentrations at low wind speeds should be estimated", this has been discussed in Section 4.3 Figure 10 panel f) page 25 lines 8-12 as "background concentrations are underestimated at the beginning of the day (1-4 UTC) and are overestimated during nighttime (19-22 UTC) in days with calm conditions."

**10. Reviewer #2.** 8.) P. 11 line 10: Does wind channelling affect the ratio ws_sfc/ws_bh?

**Authors:**

No. Wind channelling does not affect the ratio ws_sfc/ws_bh because in the formulation we consider that channelling would affect equally winds at surface and rooftop level. Hence, dividing the channelling effect by itself would give 1 and it would be omitted.

We added a note to clarify this in the revised manuscript in Section 2.3.3 page 12 lines 2-4 as follows, "wind channelling does not affect the ratio ws_sfc/ws_bh because we assume that channelling affects equally winds at surface and rooftop level".

**11. Reviewer #2.** 9.) P.12 line 5 - 10: Determination of the surface background concentrations needs more explanation. An illustration of the surface background concentration as function of building density would be helpful for understanding how it is derived from the rooftop background under different stability conditions.

**Authors:**

We added the illustration below to the revised manuscript showing the adimensional vertical mixing variable ($fac_{bg}$) that is multiplied to rooftop background to obtain surface background concentration as a function of building density and atmospheric stability.

Under low building density (i.e. *bd* below 0.1), background concentrations at surface level and over-roof level are assumed to be similar because there are almost no buildings acting as barriers. When building density increases, the difficulty of the overlying air masses to penetrate the street cavities (building height is 20 m in the illustration) is assumed to increase and more difference in concentrations is expected as a consequence. Under convective conditions, we expect more air mixing between street air and overlying air. Hence, for those conditions we assign a background within the street that is higher compared to stable atmospheric cases. Under stable atmosphere, we assume that a decrease of air mixing will increase air stratification bringing more difference in concentrations between over-roof and surface level concentrations. We include the image below in the revised manuscript in Section 2.3.3 Page 12 to visually support the explanation of the background decay method.

[Figure]

**Figure 2. Illustration of the background decay method concept. Building height is approx. 20 m.**

**12. Reviewer #2.** 10.) P.15 line 11 - 12: Which QA/QC procedure was in place for the monitoring with passive dosimeters?

**Authors:**

Duplicate dosimeters (reproducibility) were installed in some sites, and other dosimeters were installed in the permanent XVPCA network sites for comparison with reference

instrumentation.

The dosimeters were 7 cm diffusion tubes (Palm, GRADKO) that were sent to the laboratory once removed to obtain the concentrations at ambient conditions (nonstandard). Although the concentrations obtained with the dosimeters were ambient, the comparison with the data supplied by the XVPCA network permitted to correct the concentrations with the measures obtained using reference instrumentation at standard conditions. Therefore the concentrations corrected are equivalent to the standard conditions.

**13. Reviewer #2.** 11.) Table 3 and Figure 8: Measurements at station Gracia-Sant Gervasi are underestimated by all three model configurations in the daytime between morning and afternoon rush hours. Table 3 shows a positive bias for CALIOPE-urban-nl (marked as best performance for MB at this site), but this is deceiving since Figure 8 shows that overestimation at rush hours increased the bias. Obviously, the traffic increment is not correctly represented at this site. Could this be caused by the missing contribution from recirculation of traffic exhaust?

**Authors:**

The area of Gracia-Sant Gervasi site is a wide street area, which has a large street width compared to building height thus a low aspect ratio (i.e. approx. 0.38). According to Oke (1988), this kind of street is considered to be in the transition between isolated roughness and wake interference flows. The recirculation in that kind of geometrical settings is not as well documented as skimming flow cases (i.e. higher aspect ratio), where a stable recirculatory vortex is established in the canyon. Hence, we do not expect the missing contribution to be from traffic exhaust recirculation. In addition, in case of missing a relevant contribution from recirculation of traffic exhaust in this site we would expect to miss a similar contribution in all the other sites, specially in the street canyons. From the results analysis, we didn't miss that relevant contribution in all the other sites.

**14. Reviewer #2.** 12.) P.19 line 9 and Figure 8: Give the possible reason for the afternoon underestimation of $NO_2$ concentrations at sites with low traffic. The underestimation of $NO_2$ in the afternoons could also be linked to photochemical conversion. Therefore, I recommend to repeat the plots of Figure 8 for $NO_x$ concentrations.

**Authors:**

As suggested by the reviewer, we repeated the plots of Figure 8 for $NO_x$ concentrations below. We believe that the possible reason for the afternoon underestimation may be

the overestimated mixing from WRF that leads CMAQ to $NO_2$ underestimations over the city. We find very similar $NO_2$ and $NO_X$ afternoon underestimations, indicating that photochemical conversion may not be the principal reason.

[Figure]

**Figure 3. $NO_X$ average daily cycle at all sites described in Sect. 3.1 during April and May 2013 for weekday and weekend. Observations are represented in black coloured lines, red lines are CALIOPE, blue lines are CALIOPE-Urban and green lines represent CALIOPE-Urban without local developments (CALIOPE-Urban-nl).**

**15.Reviewer#2.Technical Corrections**

P. 6 line 8: "This approach addresses" fits better.

**Authors: Amended**

P.15 line 12: In every km2 grid cell?

**Authors:**

The measurements were taken independently from the model grid. Every km$^2$ in the manuscript refers to square kilometers of surface. A comment has been added in the manuscript in page 16 line 2 making it explicit: "In every km$^2$ of surface there were at least two dosimeters".

Figure 11: It should be mentioned in the figure caption whether the resolution is 10m x 10m for the entire concentration map or only in the 250m buffers along streets.

**Authors:**

We added the following note in the figure caption (page 26, Figure 12 in revised manuscript): "The resolution is for the entire concentration map."

Figure A1: What explains the zero values for the aspect ratio values in the scatterplot?

**Authors:**

The aspect ratio is assumed to tend to zero when there are no buildings on street segments sides within a distance of 100m. The algorithm to assign aspect ratio to a street segment follow this procedure:

- Build two rectangular buffers at each side of street segment given a rectangle side of 100 m (i.e. set as the maximum distance between road edge and buildings to be considered a street).
- Intersect the Barcelona buildings dataset with the two buffers
- If there are buildings at both sides:
  - Estimate the minimum distance between road edge (line) and buildings, which is assumed to represent the distance between road edge and buildings on the side of the street. Add distances at both sides of the road edge to obtain the street width.
  - Estimate the average building height of the buildings falling in both buffers
  - Estimate the aspect ratio by dividing average building height by street width.
- If there are no buildings at both sides: assign aspect ratio equal zero.

References

Beevers, S., Kitwiroon, N., Williams, M.L., and Carslaw, D.C. "One way coupling of CMAQ and a road source dispersion model for fine scale air pollution predictions". Atmospheric Environment 59, pp. 47–58. DOI: https://doi.org/10.1016/j.atmosenv.2012.05.034. 2012.

Benson, P.E., CALINE4-a Dispersion Model for Predicting Air Pollution Concentration Near Roadways. FHWA/CA/TL-84/15, p. 245. 1989.

Berkowicz, R. "A simple model for urban background pollution". Environmental Monitoring and Assessment 65.1-2, pp. 259–267. DOI : https://doi.org/10.1023/A:1006466025186. 2000.

Fagerli, H., Denby, B., and Wind, P. "Assessment of LRT contribution to cities in Europe using uEMEP?". 2019.

Heist, D., Isakov, V. Perry, S., Snyder, M., Venkatram, A."Estimating near-road pollutant dispersion: A model inter-comparison". Transportation Research Part D: Transport and Environment 25, 93-105. 2013.

Hood, C., Mackenzie, I., Stocker, J., Johnson, K., Carruthers, D., Vieno, M., and Doherty, R. "Air quality simulations for London using a coupled regional-to-local modelling system". Atmospheric Chemistry and Physics Discussions February, pp. 1–44. DOI: https://doi.org/10.5194/acp-18-11221-2018. 2018.

Hotchkiss and Harlow. "Air Pollution Transport in Street Canyons". EPA-R4-73-029. 1973.

Isakov, V. et al. "Air Quality Modeling in Support of the Near-Road Exposures and Effects of Urban Air Pollutants Study (NEXUS)". International Journal of Environmental Research and Public Health 11.9, pp. 8777–8793. DOI: https://doi.org/10.3390/ijerph110908777. 2014.

Jensen, S. S., Ketzel, M., Becker, T., Christensen, J., Brandt, J., Plejdrup, M., Winther, M., Nielsen, O.K., Hertel, O., and Ellermann, T. "High resolution multi-scale air quality modelling for all streets in Denmark". Transportation Research Part D: Transport and Environment 52, pp. 322–339. DOI: https://doi.org/10.1016/j.trd.2017.02.019. 2017.

Kim, Y., Wu, Y., Seigneur, C., and Roustan, Y. "Multi-scale modeling of urban air pollution : development and application of a Street-in-Grid model by coupling MUNICH and". Geoscientific Model Development Discussions September, pp. 1–24. DOI: https://doi.org/10.5194/gmd-11-611-2018. 2018.

Maiheu, B., Lefebvre, W., Walton, H., Dajnak, D., Janssen, S., Williams, M., Blyth, L., and Beevers, S. Improved Methodologies for NO2 Exposure Assessment in the EU. Tech. rep. 2. VITO, 2017. URL: http://ec.europa.eu/environment/air/publications/models.htm.

Moussafir, J, Olry, C, Nibart, M, Albergel, A, Armand, P, Duchenne, C, and Thobois, L. "Aircity: a very high resolution atmospheric dispersion modeling system for Paris". American Society of Mechanical Engineers, Fluids Engineering Division (Publication) FEDSM 1. DOI: https://doi.org/10.1115/FEDSM2014-

21820. 2014.

Oke, T.R. "Street design and urban canopy layer climate". Energy and Buildings 11.1-3, pp. 103–113. DOI: https://10.1016/0378-7788(88)90026-6. 1988.

Pay, M.T., Martınez, F., Guevara, M., and Baldasano, J.M. "Air quality forecasts on a kilometer-scale grid over complex Spanish terrains". Geoscientific Model Development 7.5, pp. 1979–1999. DOI: https://doi.org/10.5194/gmd-7-1979-2014. 2014.

Rotach, M. W. "Profiles of turbulence statistics in and above an urban street canyon". Atmospheric Environment 29.13, pp. 1473–1486. DOI : 10.1016/1352-2310(95)00084-C. 1995.

Salizzoni, P., Soulhac, L., and Mejean, P. "Street canyon ventilation and atmospheric turbulence". Atmospheric Environment 43.32, pp. 5056–5067. DOI : 10.1016/j.atmosenv.2009.06.045. URL: http://dx.doi.org/10.1016/j.atmosenv.2009.06.045, 2009.

Snyder, M.G., Venkatram, A., Heist, D.K., Perry, S.G., Petersen, W.B., and Isakov, V. "RLINE: a line source dispersion model for near-surface releases". Atmospheric Environment 77, pp. 748–756. DOI : https://doi.org/10.1016/j.atmosenv.2013.05.074. 2013.

Soulhac, L., Salizzoni, P., Cierco, F. X., and Perkins, R. "The model SIRANE for atmospheric urban pollutant dispersion; part I, presentation of the model". Atmospheric Environment 45.39, pp. 7379–7395. DOI: https://10.1016/j.atmosenv.2011.07.008. 2011.